

**An assessment of latest Cretaceous *Pycnodonte vesicularis* (Lamarck, 1806) shells as records for**
**palaeoseasonality: A multi-proxy investigation**
de Winter, Niels J.*[1], Vellekoop, Johan*[1,2], Vorsselmans, Robin[2], Golreihan, Asefeh[2], Soete, Jeroen[2],
Petersen, Sierra V.[3], Meyer, Kyle W.[3], Casadio, Silvio[4], Speijer, Robert P.[2], Claeys, Philippe[1]
[1]*Analytical, Environmental and Geo-Chemistry (AMGC), Vrije Universiteit Brussel (VUB), Brussels,*
*Belgium.*
[2]*Department of Earth and Environmental Science, KU Leuven, Heverlee, Belgium*
[3]*Earth and Environmental Sciences Department, University of Michigan, Ann Arbor, Michigan, USA.*
[4] *Escuela de Geología, Paleontología y Enseñanza de las Ciencias, Universidad Nacional de Río Negro,*
*CONICET, General Roca, Argentina.*
*Niels J. de Winter and Johan Vellekoop contributed equally to this work
Corresponding author: niels.de.winter@vub.be
Target journal: Climate of the Past
**Abstract**
In order to assess the potential of the honeycomb oyster *Pycnodonte vesicularis* for the
reconstruction of palaeoseasonality, several specimens recovered from the late Maastrichtian
Neuquén Basin (Argentina) were subject to a multi-proxy investigation, involving scanning
techniques, trace element and isotopic analysis. Combined CT scanning and light microscopy reveals
two major calcite micromorphologies in *P. vesicularis* shells (vesicular and foliated calcite). Micro-
XRF analysis and cathodoluminescence microscopy show that reducing pore fluids were able to
migrate through the vesicular portions of the shells (aided by bore holes) and cause recrystallization
and precipitation of secondary carbonate in the porous micromorphology, thus rendering the
vesicular portions not suitable for palaeoenvironmental reconstruction. In contrast, stable isotope
and trace element compositions show that the original chemical composition of the shell is well-
preserved in the denser, foliated portions, which can therefore be reliably used for the
reconstruction of palaeoenvironmental conditions. Stable oxygen and clumped isotope thermometry
on carbonate from the dense hinge region yield sea water temperatures of 11°C, while previous
TEX$_{86}^{H}$ palaeothermometry yielded much higher temperatures. The difference is ascribed to seasonal
bias in the growth of *P. vesicularis*, causing warm seasons to be underrepresented from the record,
and TEX$_{86}^{H}$ palaeothermometry being potentially biased towards warmer surface water
temperatures. Superimposed on this annual mean is a seasonality in δ[18]O of about 1‰, which is
ascribed to a combination of varying salinity due to fresh water input in the winter and spring season
and a moderate temperature seasonality. Attempts to independently verify the seasonality in sea
water temperature by Mg/Ca ratios of shell calcite are hampered by significant uncertainty due to
the lack of proper transfer functions for pycnodontein oysters. The multi-proxy approach employed
here enables us to differentiate between well-preserved and diagenetically altered portions of the
shells and provides an improved methodology for reconstructing palaeoenvironmental conditions in
deep time. While establishing a chronology for these shells was severely complicated by growth



cessations and diagenesis, cyclicity in trace elements and stable isotopes allowed a tentative
interpretation of the potential annual seasonal cycle in the late Maastrichtian palaeoenvironment of
the Neuquén basin. Future studies of fossil ostreid bivalves should target dense foliated calcite
rather than sampling bulk or vesicular calcite. Successful application of clumped isotope
thermometry on fossil bivalve calcite in this study indicates that temperature seasonality in fossil
ostreid bivalves may be constrained by the sequential analysis of well-preserved foliated calcite
samples using this method.

## 1. Introduction

The Late Cretaceous is generally considered a greenhouse world (e.g. Hay, 2008). Indeed,
reconstructed global mean temperatures and atmospheric $pCO_2$ concentrations for this period
generally exceed those of the present-day climate (e.g., Berner, 1990; Andrews et al., 1995; Ekart et
al., 1999; Hunter et al, 2008; Quan et al., 2009; Wang et al., 2013). As such, the Late Cretaceous may
be considered an analogue for climate in the near future if anthropogenic greenhouse gas emissions
continue unabated (IPCC, 2014; Hay, 2013; Dlugokencky, 2017). Many studies have yielded
reconstructions of Late Cretaceous climates using either climate models or a variety of proxies in
temporally long archives, such as deep-sea cores and continental sections (Pearson et al., 2001;
Huber et al., 2002; Otto-Bliesner et al., 2002; Miller et al., 2003; Friedrich et al., 2012; de Winter et
al., 2014; Vellekoop et al., 2016). Yet, although most deep time climate reconstructions so far have
focused on reconstructing mean annual temperatures (MAT), climate change also involves changes
in other climate parameters, such as precipitation, seasonality and the frequency of extreme
weather events, which all take place on timescales shorter than those that can be resolved in the
above mentioned long archives. Therefore, it is important that these climate variations are
understood on a shorter timescale.
One way to achieve such high-resolution palaeoclimate and palaeoenvironmental reconstructions is
by using marine organisms that form shells that grow incrementally. Marine bivalve shells are
excellent palaeoclimate recorders, and the relationship between their shell chemistry and the
environmental conditions in which they grow has been studied intensively (Jones, 1983; Dettman
and Lohmann, 1993; Steuber, 1996; Gillikin et al., 2005a; Elliot et al., 2009). Many geochemical
proxies have been described based on bivalve calcite. Examples include temperature calibrations for
Mg/Ca and stable oxygen isotope ratios ($\delta^{18}O$; e.g. Klein et al., 1996a; Richardson et al., 2004; Freitas
et al., 2008; Wanamaker et al., 2008), tentative salinity calibrations using Sr/Ca and the combination
of Mg/Ca and $\delta^{18}O$ (Dodd and Crisp, 1982; Klein et al., 1996a; Watanabe et al., 2001) and proxies for
palaeoproductivity, such as Ba/Ca and Mn/Ca (Lazareth et al., 2003; Gillikin et al., 2008).
Despite their potential for high-resolution palaeoenvironmental reconstruction, bivalve records
feature rarely in long timescale reconstructions (e.g. Steuber, 2005; Harzhauser et al., 2011;
Hallmann et al., 2013). A caveat in the use of bivalve records for long-term palaeoclimate
reconstructions is the potential problems that arise when using multiple bivalve species for
palaeoclimate reconstruction (Gillikin et al., 2005a; b; de Winter et al., 2017a). Culture experiments
in extant bivalve species have shown that palaeoenvironmental proxies in bivalve calcite may be
affected by internal mechanisms that are independent of the environment of the animal and are
controlled by parameters such as growth, reproductive cycle and metabolism (the so-called "vital
effects"; Dunbar and Wefer, 1984; Weiner and Dove, 2003; Gillikin et al., 2005b; Lorrain et al., 2005;
Carré et al., 2005). Such internal factors are often species-specific and limit the applicability of proxy
transfer functions from modern culture studies to multiple species in the same study or on species





for which no culture study data is available. The integration of different species of bivalves in
palaeoclimate studies is further complicated by the various ecological niches these species of
bivalves occupy, which results in great variability between their direct environments (Chauvaud et
al., 2005; Dreier et al., 2014). In addition, bivalves are mostly restricted to shallow marine and
estuarine environments. This further complicates the interpretation of bivalve records (e.g. Surge et
al. 2001, Richardson et al., 2004; Gillikin et al., 2008; Wisshak et al., 2009; Ullmann et al., 2010), as
these environments are often characterized by large variations in temperature, salinity and water
chemistry, which makes it hard to disentangle the effect of different environmental parameters on
geochemical proxies (e.g. Duinker et al., 1982; Morrison et al., 1998; Pennington et al., 2000).
The above-mentioned problem of combining different high-resolution climate records to study
climatic variations on a geological timescale can be overcome by combining results from multiple
well-preserved bivalve specimens of the same species and in the same geological setting. Several
studies have tried such a multi-specimen approach to trace changes in high-resolution climate
parameters, such as seasonal variations, over geological timescales (Dettman and Lohmann, 2000;
Dettman et al., 2001; Steuber, 2005; Gutiérrez-Zugasti et al., 2016). However, such reconstructions
require bivalve species that preserve well, are geographically widespread, have a high occurrence
frequency over longer timescales and record seasonal-scale variations within their shell.
Potential candidate species are bivalves of the genus *Pycnodonte*. This genus of oysters (Bivalvia:
Ostreoida; Fisher von Waldheim, 1835) is characterized by a well-developed commissural shelf and a
vesicular shell structure (hence the name "honeycomb oyster" or "foam oyster"; Stenzel, 1971;
Hayami and Kase, 1992). Members of the genus *Pycnodonte* are found in geological deposits from
the Lower Cretaceous to the Pleistocene. The appearance of *Pycnodonte* shells in a wide range of
palaeolatitudes and geological settings, especially in the Cretaceous, makes them a promising
archive for high-resolution climate reconstruction (Ayyasami, 2006; Fossilworks, 2017). As
mentioned in Titschak et al. (2010), records from large and long-living bivalves, such as *Pycnodonte*,
provide several advantages in comparison with other seasonality archives. They are slow-growing,
reducing kinetic effects and disequilibrium fractionation of stable isotopes (McConnaughey 1989;
Abele et al., 2009). In addition, *Pycnodonte* bivalves likely did not have symbionts, in contrast to, for
example, Tridacnid bivalves (Elliot et al., 2009). This means that *Pycnodonte* bivalves take up
nutrients and other elements directly from their environment, simplifying the interpretation of their
shell composition. Their low-Mg calcite shells are less prone to diagenetic alteration than shells
made of aragonite or high-Mg calcite (Al-aasm and Veizer, 1986; Pirrie and Marshall, 1990), and their
sedentary life mode ensures that they fossilize in life position. The latter enables the integration of
environmental information extracted from the sediments in which they are fossilized into the
discussion of their shell chemistry.
The species *Pycnodonte vesicularis* (Lamarck, 1806) is one of the most common and long-ranging
species of *Pycnodonte.* Therefore, in this study the potential for *P. vesicularis* to be used as a record
for sub-annual environmental variability in the Late Cretaceous is explored. The present study
focuses on the characteristics of fossil specimens of *P. vesicularis* from the upper Maastrichtian
Jagüel Formation of the Bajada de Jagüel section, Argentina (**Figure 1A**). A range of qualitative, semi-
quantitative and quantitative methods are applied to investigate the nature of the *P. vesicularis* shell
material, shell morphology and its preservation state. The aim of this multi-proxy approach is to
characterize the structure and chemical composition of the *P. vesicularis* shell and its development
through the lifetime of the animal and to assess its potential as a recorder of paleoseasonality.



**2. The species *Pycnodonte vesicularis***
*Pycnodonte vesicularis* was reclining and inhabited muddy bottoms on the shallow marine shelf with
a low sedimentation rate (e.g. Brezina et al., 2014). The individual variability is very extensive in *P.*
*vesicularis*, involving, among others, the outline of valves, their convexity, the thickness of the walls,
the dimensions, the deepness, shape and position of the adductor muscle scar, as well as the
characteristics of chomata (Pugaczewska, 1977; Brezina et al., 2014). This variability depends on the
age of the individual and on local environmental conditions, especially on the character and grain
size of the substrate. According to Berzina et al. (2014), about one third of *P. vesicularis* valves at
Bajada de Jagüel are mature (gerontic) specimens, characterized by relatively thick valves (>10mm)
with a well-developed vesicular layer. Given their longer life span, such mature specimens of *P.*
*vesicularis* were considered most suitable for the present investigation.
In the past, several studies have made an attempt to calculate the age of individuals of *P. vesicularis*
based on the number of laminae in the complex of lamellar and vesicular layers (Nestler, 1965), or
the number of growth lines of the ligament (Müller, 1970). Yet, so far no studies have investigated
the potential of *P. vesicularis* shells as palaeoseasonality records based on their geochemical
signature. Given the species-specific relationships between environmental parameters and bivalve
shell geochemistry, in an ideal situation, a culture experiment would be used to determine these
relationships for *Pycnodonte* bivalves. Unfortunately, no extant species of the genus *Pycnodonte* are
known, rendering culture experiments for these species impossible. However, two species of the
closely related pycnodontein genus *Neopycnodonte* (Stenzel, 1971) are found in deep-sea habitats
today (*Neopycnodonte conchlear*, Poli, 1795, and *Neopycnodonte zibrowii*; Videt, 2004; Wisshak et
al., 2009), whereas the extant pycnodontein genus *Hyotissa* is characterized by a shallow-marine
distribution (Titschack et al., 2010). Detailed studies of the shell morphology and chemical
composition of *N. zibrowii* and *Hyotissa hyotis* are reported in Wisshak et al., (2009) and Titschak et
al., (2010), respectively, and can be used as a basis for comparison of the *Pycnodonte* oyster shells.

**3. Geological Background**
**3.1 Paleogeographical context**
The studied specimens were collected from the Bajada de Jagüel (BJ) section (38°06'10.5"S,
68°23'20.5"W). The site is situated in the Neuquén Basin in Argentina. The Neuquén Basin is
bordered to the south by the North Patagonian massif and to the northeast by the Sierra Pintada
massif (**Figure 1B and 1C**). The Bajada de Jagüel section has a palaeolatitude of ~43˚S ± 2˚relative to
the palaeomagnetic reference frame of Torsvik et al. (2012) according to palaeolatitude.org (van
Hinsbergen et al. 2015). A large transgression from the South Atlantic into the basin (Bertels, 2013)
occurred from the late Maastrichtian to early Danian, during a time of relative tectonic quiescence
and low magmatic activity (Malumian et al., 2011).
**3.2 Palaeoenvironment**
The Maastrichtian mudstones of the Jagüel Fm are homogeneous and intensely bioturbated,
indicating a well-oxygenated seafloor, with palaeodepths of approximately 50-75 m (Scasso et al.,
2005; Woelders et al., 2017; see also **Figure 1**). A coarse-grained, mottled, clayey sandstone bed, 15-
25 cm thick, separates the Maastrichtian and Danian mudstones. This sandstone bed represents the
K-Pg boundary and is interpreted to have resulted from a tsunami wave, related to the Chicxulub
impact event (Scasso et al. 2005). During the late Maastrichtian and early Danian, North and Central



Patagonia experienced a warm, humid climate. Pollen records suggest rainforests, coastal mangrove
forests and swamp communities in the region (Baldoni 1992; Kiessling et al. 2005; Barreda and
Palazzesi 2007; Iglesias et al. 2007; Palazzesi and Barreda 2007). This vegetation type is classified as
megathermal and indicates average air temperature of 24°C or higher (Barreda and Palazzesi 2007;
Palazzesi and Barreda 2007; Barreda et al. 2012). Average annual sea surface temperatures are
estimated to have been 26-29°C in the latest Maastrichtian at Bajada de Jagüel, based on $TEX_{86}^{H}$-
palaeothermometry (Woelders et al., 2017; **Figure 1D**). While hypersaline conditions have been
inferred for the northernmost part of the Neuquén Basin, the central part of the Neuquén Basin,
where the BJ site is located, is suggested to have experienced more normal marine conditions. The
latter is evidenced by the presence of planktic foraminifera, dinocysts and relatively few terrestrial
palynomorphs (Prámparo et al. 1996; Prámparo and Papú 2006; Woelders et al. 2017). Yet,
Woelders et al. (2017) inferred enhanced runoff and stratification of the water column at the Bajada
de Jagüel site during the late Maastrichtian warming (450-150 kyr before the K-Pg boundary). Hence,
salinity may have deviated from normal marine during the lifetime of the *Pycnodonte* specimens
studied here.

**4. Materials and methods**
**4.1 Sample acquisition and preparation**
Seven specimens of *Pycnodonte vesicularis* were collected from the upper Maastrichtian Jagüel
Formation in the Bajada de Jagüel section (**Figure 1**), labelled "M0", "M4", "M5", "M6", "M8",
"M10" and "M11". All shells were collected from the upper 5m of below the Cretaceous-Palaeogene
boundary (see **Figure 1**). Four of these specimens ("M0", "M4", "M6" and "M11", see **Figure 2**)
represent completely preserved left valves of mature specimens of *P. vesicularis* (c.f. Pugaczewska,
1977), while the remaining three ("M5", "M8" and "M10") were incomplete. Specimens were
selected that differ from each other in morphology, body size and extent of biodegradation, to
assess both the potentials and possible pitfalls of this taxon as a palaeoseasonality recorder. The
four complete shells were cleaned and cast into Araldite® 2020 epoxy resin (Araldite, Basel,
Switzerland) before being cut along the major growth axis of the shell using a slow rotating rotary
saw (Ø 1 mm). A parallel slab was cut out of one half of the shell, while the other half was preserved
(archive half). The resulting thick section, with a typical thickness of 4 mm, was polished using a
series of progressively higher-grade silicon carbide polishing disks (up to P2400) to allow a smooth
surface for sampling and imaging. The remaining three shells were left untreated and were only used
for bulk analysis.
**4.2 Colour scanning and microscopy**
Polished surfaces of shell sections were colour-scanned at 6400 dpi resolution (~4 µm resolution)
using an Epson 1850 flatbed scanner. Shell structures were studied and imaged at 50x magnification
using an Olympus BX60 optical microscope (KU Leuven, Belgium). In order to study the preservation
of pristine calcite in the *P. vesicularis* shells, shell slabs were studied using cathodoluminescence
microscopy using a Technosyn Cold Cathodoluminescence model 8200, mark II microscope operated
at 16-20 kV electron gun potential, 420 µA beam current, 0.05 Torr (6.6 * $10^{-5}$ bar) vacuum and 5
mm beam width (KU Leuven, Belgium). Cathodoluminescence (CL) refers to the emission of light
from material during excitation by an electron beam. The wavelength (i.e. colour) of the emitted
light depends on the crystal lattice structure and on activators, i.e. light emitting centres constituted
by chemical elements or crystal defects. CL microscopic observations of the shell sections thus



enable the recognition of crystal defects and to evaluate the preservation state of the samples (e.g.
overgrowth, recrystallisation, dissolution). They allow to assess to what extent the obtained element
concentrations and isotopic ratios reflect the original shell signature (Barbin, 2000).
**4.3 Porosity and trace element analysis**
In order to visualize shell structure and the pore network, high-resolution 3D micro-tomography
analysis was carried out on the archive half of *Pycnodonte* specimens using a General Electric
Nanotom microCT X-Ray CT scanner (KU Leuven, Belgium). One entire shell halve was scanned at a
30 μm spatial resolution while representative shell pieces of interest were scanned at 1.5 μm
resolution. The CT images were segmented in Matlab by applying a dual thresholding algorithm. The
shell porosity was rendered in 3D and labelled in Avizo Fire 7.0. Pore parameters were calculated in
Avizo and Matlab. Micro-XRF measurements were carried out using a Bruker M4 Tornado micro-XRF
scanner at the XRF platform of the Analytical, Environmental and Geochemistry group at the Vrije
Universiteit Brussel (Brussels, Belgium). Details on the setup of the M4 Tornado μXRF scanner can be
found in de Winter and Claeys (2016). μXRF mapping was done using the M4 Tornado's Rh-anode X-
Ray tube under maximum source energy settings (50kV, 600 μA) using two silicon drift detectors, a
spatial pixel resolution of 50 μm and an integration time of 1 ms per pixel. μXRF line scans of the
hinges of shells M0, M4, M6 and M11 were measured on the M4 Tornado in point-by-point mode
(see de Winter et al., 2017a) using maximum source energy settings (50kV, 600 μA), a spot size of 25
μm, a spatial sampling resolution of 50 μm and an integration time per point of 60 seconds (1085
measurements in total). This measurement strategy allowed XRF spectra to accumulate enough
counts to reach the Time of Stable Reproducibility and Accuracy (de Winter et al., 2017b). Line scans
were carried out in growth direction on polished cross sections through the hinge of the four *P.
vesicularis* shells (see **Figure 1**). Care was taken to limit sampling to the dense calcite in the hinge of
the shells, though observations of the microstructure of the shell hinge show that incorporation of
vesicular calcite into the profile could not always be avoided (see section 4.1.1 and 4.1.3).
**4.4 Trace elements in bivalves**
The use of trace element concentrations in fossil bivalve shells as a means of reconstructing
palaeoenvironmental conditions is subject to ongoing debate. As briefly mentioned above, some
tentative calibrations have been made that link trace element ratios in shell carbonate to
environmental conditions in modern bivalves (e.g. Jones et al., 1980; Klein et al., 1996a; Freitas et
al., 2005; Wanamaker et al., 2008). However, the degree by which the incorporation of these trace
element concentrations is controlled by the shell's environment, as opposed to internal mechanisms
(vital effects), is often uncertain (e.g. Weiner and Dove, 2003; Lorrain et al., 2005; Gillikin et al.,
2005b). An example of this is the Mg/Ca ratio, which is widely thought to reflect the calcification
temperature of the shell (e.g. Klein et al., 1996a). While the Mg/Ca palaeothermometer is commonly
applied in foraminifera studies (e.g. Ederfield and Ganssen, 2000; Lear et al., 2000), calibrations of
this proxy for different bivalve taxa vary widely (Klein et al., 1996a; Vander Putten et al., 2000;
Takesue and van Geen, 2004; Freitas et al., 2005; Wanamaker et al., 2008; Surge and Lohmann,
2008; Mouchi et al., 2013; see also de Winter et al., 2017a). Even Mg/Ca calibration curves for oyster
species within the same genus (*Crassostrea virginica* and *Crassostrea gigas*; Surge and Lohmann,
2008 and Mouchi et al., 2013, respectively) yield very different results, illustrating that the
temperature dependence of Mg/Ca ratios in bivalve calcite is not straightforward. Furthermore, it
has been shown that the incorporation of Mg (and other trace elements, such as Sr and Mn) into
bivalve shells does not happen in equilibrium with ambient concentrations (Weiner and Dove, 2003).
Relationships of bivalve Mg/Ca ratios with temperature are also known to break down during
periods of growth stress (Lorens and Bender, 1980; Takesue and van Geen, 2004). Some Mg in
bivalve shells is associated with organic molecules in the matrix in the shell rather than being




substituted for Ca in the crystals of bivalve calcite (Lorens and Bender, 1980). Hence, factors
determining elemental incorporation in bivalve carbonate are partly controlled by physiological
processes and are therefore species specific (e.g. Freitas et al., 2006; 2008).
Another commonly reported ratio, that of Sr/Ca, has yielded good correlations with water
temperature for some bivalve taxa (e.g. Freitas et al., 2005), while others have shown that it strongly
covaries with changes in growth and metabolic rate in a range of taxa (Klein et al., 1996b; Lorrain et
al., 2005; Gillikin et al., 2005b). The above shows that the extent of vital effects is highly taxon-
specific and that climate reconstructions based on trace element records in bivalve shells need to be
interpreted with great care. Beside sea water temperature, attempts have been made to reconstruct
other environmental parameters, such as redox conditions and palaeoproductivity, based on trace
element records in bivalves. Examples of such proxies include elements that are enriched in
skeletons of primary producers such as Ba (Gillikin et al., 2008; Marali et al., 2017), redox-sensitive
elements like Mn (Freitas et al., 2006) and micronutrients such as Zn and Cd, which are known to be
taken up into bivalve shells and whose concentration profiles reflect changes in palaeoproductivity
(Carriker et al., 1980a; Calmano et al., 1993; Jackson et al., 1993; Wang and Fisher, 1996; Guo et al.,
1997). It has been demonstrated that seasonal records of these proxies are reproducible between
different shells in the same environment (Gillikin et al., 2008). While these proxies have not been
explored in detail, their interpretation gives additional information about the ambient sea water
chemistry and illustrates the advantage of applying the multi-proxy approach to reconstruct
seasonality from bivalve shells (de Winter et al., 2017a).

**4.5 Stable isotope analyses**
Samples for stable isotope analysis were drilled using a microscope-guided Merchantek drill, coupled
to Leica GZ6 microscope, equipped with a 300 µm diameter tungsten carbide drill bit (AMGC group,
VUB, Belgium). Spatial sample resolutions smaller than the diameter of the drill were obtained by
abrading consecutive samples off the side of the sampling front. This was achieved by moving in
steps of 100µm along a ±2 mm wide linear sampling path, oriented parallel to the growth lines of the
shell and in the growth direction of the shell (447 measurements in total; see also Van Rampelbergh
et al., 2014; Vansteenberge et al., **in review CHEMGEO**). Dense foliated calcite in the hinge of the
shells was targeted in sampling for stable isotope analysis, but as a result of the shell structure (see
discussion below) the incorporation of vesicular calcite could not always be excluded. Note that, as a
consequence of the abrading sampling strategy, the width of the sampling path for IRMS samples is
much larger (2 mm) than the width of the sampling path of a µXRF line scan (25 µm). This caused
more vesicular calcite to be incorporated into stable isotope measurements than in µXRF
measurements, as it was easier to avoid the vesicular structure in the µXRF measurements. Aliquots
of ±50 µg of sampled calcite were allowed to react with 104% phosphoric acid (H$_3$PO$_4$) at 70°C in a
NuCarb carbonate preparation device and stable oxygen and carbon isotope ratios ($\delta^{13}$C and $\delta^{18}$O)
were measured using a NuPerspective Isotope Ratio Mass Spectrometer (AMGC group, VUB,
Belgium). Analytical uncertainty was determined by repeated measurement (N = 110) of the in-
house reference material MAR2 (Marbella marble, $\delta^{13}$C: 3.41 ± 0.10 ‰VPDB; $\delta^{18}$O: 0.13 ± 0.20
‰VPDB; 1 standard deviation, SD) and found to be 0.02‰ and 0.08‰ for $\delta^{13}$C and $\delta^{18}$O values (1
SD), respectively. This MAR2 reference material was previously calibrated using the international
NBS-19 stable isotope standard (Friedman et al., 1982). All stable isotope values are reported in
permil relative to the Vienna Pee Dee Belemnite standard (‰VPDB). While µXRF and IRMS
measurements were carried out on the same transect, small differences in the length of the records
did occur and these were corrected by linearly rescaling the stable isotope records to match the
length of trace element records in the same shell.





### 4.6 Clumped isotope analysis

**4.6 Clumped isotope analysis**
The stable and clumped isotopic composition of five shells (M4, M5, M8, M10 and M11) was
measured at the University of Michigan Stable Isotope Laboratory. Bulk sampling for clumped
isotope analysis was carried out in two ways: 1) Of three shells (M5, M8 and M10), slabs of dense
calcite were broken off the ventral margin and powdered by hand. 2) Of four shells (M4, M5, M8 and
M11), samples were drilled from the dense hinge area. Sample preparation was performed on a
manual extraction line following Defliese et al. (2015), with the temperature of the Porapak$^{TM}$ trap
increased to avoid fractionating stable isotope values (Petersen et al., 2016). Aliquots of 3.5-5 mg
carbonate powder were reacted with phosphoric acid ($H_3PO_4$) at 75°C and sample $CO_2$ was analysed
on a ThermoFinnegan MAT253 equipped with Faraday cups to measure m/z 44-49. Each sample was
analysed for 5 acquisitions of 12 cycles each and calibrated relative to heated (1000°C) and $H_2O$-
equilibrated (25°C) gas standards and two in-house carbonate standards (Carrara Marble and
Aragonitic Bahamanian Ooids). Gas standards were used to convert unknowns into the absolute
reference frame (Dennis et al., 2011) and carbonate standards (Carrara Marble and Aragonitic
Bahamanian Ooids) were used to quantify reproducibility of reacted samples. $\delta^{18}O_{water}$ values were
calculated using the calcite-$H_2O$ equation of Kim and O'Neil (1997). External (long term) error on the
$\Delta_{47}$ value was found to be 0.011‰ (1σ), based on companion measurements of carbonate standards
(see **supplementary data 1**). Data presented in the main manuscript were processed using the
Santrock/Gonfiantini parameters and the high-temperature composite calibration of Defliese et al.
(2015). Further details on the measurement and calibration procedure of clumped isotope
thermometry are found in **supplementary data 1**, along with raw data processed using both
Santrock/Gonfiantini and Brand parameters.

**5. Results**
**5.1 *Pycnodonte vesicularis* shell structure**
**5.1.1 Shell microstructures**
An overview of the results of colour scanning, microscopic analyses and µXRF mapping on one of the
*P. vesicularis* specimens (M11) reveals the structure of the shells of these honeycomb oysters (**Figure
3; supplementary data 3**). A cross section through the shell in direction of maximum growth (**Figure
3A**) reveals a layered shell structure with laterally continuous growth increments similar to those
found in modern ostreid shells (e.g. Carriker et al., 1980b; Surge and Lohmann, 2008; Ullmann et al.,
2013). Growth increments are characterized by an alternation of dense, foliated calcite layers with
lighter coloured, more porous, vesicular ("chalky") calcite layers that are characteristic for the family
Gryphaeidae (Linnaeus, 1758; Carriker et al., 1980b; Bieler et al., 2004; Surge and Lohmann, 2008).
The porosity in these vesicular layers is visualized in microscopic images (**Figure 3C-E**). Microscopic
images also show that the hinge of the shell is mostly devoid of this vesicular structure, and instead
consists of a close packing of foliated calcite layers (**Figure 3A** and **Figure 3G**). However, in parts of
the hinge small layers of vesicular calcite are also visible between the foliated layers in **Figure 3G**. In
places where these vesicular layers are interlocked between foliated layers, the transition between
the two microstructures is gradual. Microscopic images (**Figure 3D-E**) show that farther away from
the shell hinge, the transitions between foliated calcite and vesicular calcite are sharp, and that
individual layers of foliated and chalky calcite can be very thin (<30 µm; **Figure 3D**). Pores in the
vesicular calcite are heterogeneous in size and can be up to 200 µm wide. While the shell structure is
in general very well preserved (**Figure 3C-G**), it is disturbed in some areas by patches of different



texture, or holes that have been previously ascribed to boring polychaete worms (Brezina et al.,
360    2014).

**5.1.2 Porosity**

Micro-CT images of one of the *P. vesicularis* specimens (M4) further illustrate the distribution of
porosity in the shell (**Figure 4**). Porosity analysis based on micro-CT scanning confirms the
microscopic observations of porous vesicular calcite and dense foliated calcite layers in the shells.
Quantitative analyses of porosity through the shell (porosity logs) on the high-resolution CT scan of a
small part of the shell (**Figure 4B**) shows that the distribution of porosity strictly relates to growth
layering of the shell. The porosity log perpendicular to the growth layering (**Figure 4E**) shows that
porosity is almost absent in the foliated calcite layers and reaches up to 65% of the shell volume in
the most porous vesicular layers. Total shell CT scan results reveal that the average porosity in the
shell is 21%. Results of CT scanning and microscopy show that, while the calcite in the vesicular
microstructure was affected by diagenesis, the original porosity in these *P. vesicularis* shells has been
preserved almost completely, and the filling of pores by recrystallized calcite is relatively uncommon
(see **Figure 3D-E**).

**5.1.3 Chemical heterogeneity and cathodoluminescence**

Heterogeneity in the *P. vesicularis* shell is also evidenced by the distribution of iron (Fe) and
manganese (Mn) in the shell, as illustrated by µXRF mapping (**Figure 3B**). The map shows that the
vesicular layers in the shell are characterized by higher concentrations of Fe and Mn than the dense
foliated calcite layers. Parts of the shell that were perforated by bore holes have especially high
concentrations of Fe and Mn, and these holes are surrounded by a corona of elevated Fe and Mn
concentrations (**Figure 3H and I**). A close-up of the shell hinge in **Figure 3B** confirms that it consist
almost entirely of dense foliated calcite with low Fe and Mn concentrations. The same close-up also
illustrates the limitations of µXRF mapping with a spot size of 25 µm. The method is not able to
resolve variations in the concentration of Fe and Mn on the scale of fine (<30 µm) laminations in the
shell hinge. A composite of cathodoluminescence microscopy images of the same area (insert in
**Figure 3A**) complements µXRF mapping by showing in more detail that the foliated calcite of the
shell hinge is characterized by microscopic growth increments that show a dull luminescence. Only
the largest increments can be distinguished on the µXRF map. In calcite, $Mn^{2+}$ is the main
luminescence activator causing emission of yellow to orange light (~620 nm; Machel and Burton,
1991) of which the intensity is positively correlated with the Mn concentration (de Lartaud et al.,
2000a; Habermann, 2002; Langlet et al., 2006; de Winter and Claeys, 2016). Indeed, brighter layers
in the CL image correspond with higher Mn values in the XRF map. An enlarged version of the CL
composite shown in **Figure 3** is given in **supplementary data 2** and XRF Mn and Fe maps of all shells
are given in **supplementary data 3**.

**5.2 Trace element profiles**

Results of XRF line scans through all *P. vesicularis* shells featuring in this study are given in
**supplementary data 4**. Quantitative XRF line scans through the hinge of the *P. vesicularis* shells yield
records of [Ca], [Si], S/Ca, Zn/Ca, Sr/Ca, Mg/Ca, [Mn] and [Fe] in growth direction through the dense
hinge area of the shells (**Figure 5**). All measured XRF data is directly represented in **Figure 5**, only the
Mg/Ca record is plotted with a three point running average. This running average smoothes out the
variation between individual Mg/Ca measurements, because Mg is slightly more susceptible to
interferences on the XRF spectrum, causing noise on the Mg/Ca record . This results from the fact
that Mg is on the edge of the spectrum of elements measurable by the M4 Tornado µXRF scanner



and is therefore(see de Winter and Claeys, 2016; de Winter et al., 2017b). A plot of these results
shows that concentrations of calcium (Ca) and silicon (Si) in all shells generally remain above 38
mass% and below 0.5 mass%, respectively. In three out of four specimens (M0, M4 and M6),
absolute concentrations of Fe and Mn rarely exceed 800 µg/g (**Figure 5**). The exception is the iron
record of specimen M11, which shows maxima often exceeding 2000 µg/g. Fe concentrations in M6
are also elevated in comparison with M0 and M4, leading to the suggestion that there might be a
link between the presence of bore holes (observed in M6 and M11) and elevated Fe-concentrations.
A cross plot in **Figure 6A** shows that the concentrations of Fe and Mn are weakly correlated in XRF
line scan measurements. Furthermore, samples with elevated concentrations of Mn generally have
lower concentrations of Sr, especially when Mn concentrations are higher than 800 µg/g (**Figure 6B**).
Both are a sign of diagenetic alteration because Mn and Fe have been shown to be preferentially
enriched in recrystallized shell carbonates, while Sr is preferentially removed during the
recrystallisation process (Brand and Veizer, 1980; Al-Aasm and Veizer, 1986). Trace element profiles
through the four *P. vesicularis* specimens show that there is a good overall agreement between
shells both in terms of absolute concentration of magnesium (Mg), strontium (Sr), zinc (Zn) and
sulphur (S) and their internal variation. Records of ratios of Mg/Ca, Sr/Ca, Zn/Ca and S/Ca show
quasi-cyclic oscillations. In records of Mg/Ca and Sr/Ca, these oscillations are quasi sinusoidal, while
records of Zn/Ca and S/Ca are characterized by short-lived increases relative to a baseline value.
Trace element ratios generally oscillate around a stable baseline value, though in some cases (e.g.
Sr/Ca and Mg/Ca in M11) there is a slight evolution of this baseline value in the direction of growth.
**5.3 Stable isotope analysis**
**5.3.1 Stable isotope records**
Records of stable oxygen isotope ratios ($\delta^{18}O$) and stable carbon isotope ratios ($\delta^{13}C$) are plotted
together with trace element ratios in **Figure 5**. As in the trace element records, absolute values as
well as internal variation of stable isotope records show good agreement between shells. Values in
the $\delta^{18}O$ record oscillate around a baseline value of -1.5‰. The $\delta^{13}C$ baseline values are a bit more
variable, possibly showing a late ontogenetic trend in M6, but remaining stable at 2‰ in the other
specimens. Stable oxygen isotope ratios remain between -2.5‰ and -0.5‰ for the majority of the
records, exceptions being $\delta^{18}O$ values below -3‰ in a few measurements in M4, the central part of
the M6 record, and a few measurements in the youngest part of the M0 record. Similarly, $\delta^{13}C$ ratios
in all shells remain between 1.5‰ and 3.5‰, except for the latter cases. Cross plots between
isotope ratios show that samples with exceptionally low $\delta^{18}O$ values (<-3‰) often also exhibit
decreased $\delta^{13}C$ values (<1.5‰; **Figure 6D**). This relationship between $\delta^{18}O$ and $\delta^{13}C$ values is
significant in shells M4, M6 and M11, and not in M0. Such a relationship between $\delta^{18}O$ and $\delta^{13}C$ has
often been interpreted as a sign of diagenetic alteration. Therefore, the absence of this relationship
in M0 in contrast to the other shells shows that the stable isotope profile from the hinge of shell M0
is least affected by diagenetic alteration. Stable oxygen and carbon isotope records seem to show
quasi-periodic variations around these baseline values, with amplitudes of about 1‰ and 0.5‰
respectively (**Figure 5**). Cross plots of proxy records show that $\delta^{18}O$ and $\delta^{13}C$ values are generally
lower in samples with elevated concentrations of Mn and Fe (**Figure 6A and 6C**).
**5.3.2 Clumped isotope analysis**
Clumped isotope analyses of ventral margin calcite from three *P. vesicularis* shells from the same
locality (M5, M8 and M10) yielded $\Delta_{47}$ values of 0.699 to 0.707‰, equivalent to a temperature range
of 21-25°C using the high temperature composite calibration of Defliese et al., 2015 (see **Table 1**).
Both reconstructed temperatures and $\delta^{18}O_{seawater}$ values varied significantly between these samples,





with $\delta^{18}O_{seawater}$ ranging from -0.6‰ in M10 to -2.2‰ and -5.9‰ in M5 and M8 respectively, likely
indicating the influence of altered calcite material. This is supported by shell $\delta^{18}O$ values, which
deviate to very low values (-4‰ to -7‰VPDB in M5 and M8) well outside of the range of samples
micromilled from the well-preserved hinge carbonate (**Figure 5**). The same samples (M5 and M8)
also show relatively decreased $\delta^{13}C$ values (<1‰), further indicating that these decreased stable
isotope ratios are likely indicative of diagenetic alteration. In comparison, samples of the dense
hinge calcite from M4, M5, M8 and M11, yielded $\delta^{18}O_{seawater}$ values ranging from -1.8‰ to -2.5‰
and $\Delta_{47}$ values of 0.725 to 0.746‰, corresponding to much cooler temperatures of 9-15°C. Shell $\delta^{13}C$
and $\delta^{18}O$ values from bulk samples of hinge carbonate resemble values measured in the high-
resolution transects, further supporting the good preservation of carbonate in this area.

**6. Discussion**
**6.1 Shell preservation**
**6.1.1 Visualization of diagenesis**
The preservation of fine shell porosity measured by CT-scanning shows that if any recrystallization
occurred in the shells, it was not so extensive that the pores in the vesicular layers were filled by
secondary calcite. Yet, identifying diagenesis in *P. vesicularis* shells cannot be done based on simple
visual inspection alone. Recrystallized calcite is often characterized by elevated concentrations of
Mn and Fe, which are released into pore waters of the sediment surrounding the shell under
reducing conditions (Al-Aasm and Veizer, 1986). This allows the distribution of Fe and Mn
concentrations in the shells to be used as an indicator for the amount of recrystallization and the
preservation of the shell. The map in **Figure 3B** shows that such recrystallization is predominantly
observed in the vesicular calcite and that Fe and Mn concentrations in foliated calcite layers are low.
Coronas of elevated Fe and Mn concentrations around the bore holes in the shells confirm that
increased concentrations of Mn and Fe are leached into the shell through these holes as penetrating
pore fluid carrying these ions can more easily infiltrate the vesicular calcite layers than the foliated
calcite. The fact that shells M6 and M11, which contain the most bore holes (see **Figure 2**), have the
highest Mn and Fe values (**Figure 5**) supports this hypothesis. This pattern is confirmed by the
cathodoluminescence microscopy images, which show minimal dull luminescence in the foliated
calcite, indicative of limited contamination of the calcite by Mn and Fe (Barbin, 2000). Thin lamina
between foliated calcite layers show brighter luminescence, associated with higher concentrations
of Fe and Mn. This is in agreement with peaks in Mn and Fe observed in the µXRF profiles of M11
(**Figure 5**). Microscopic images of the foliated calcite structure (e.g. **Figure 3F-G**) further show that
the elongated crystal structure characteristic of pristine foliated shell calcite has not been
compromised by diagenesis (Ullmann et al., 2010). Comparison between the CL composite and the
µXRF map shows that, while µXRF mapping does pick up large scale diagenetic features in the shell,
it fails to reveal most of the small layers intercalated between foliated calcite layers in the shell
hinge because they are smaller than the spot size of the µXRF scanner (25 µm). This illustrates that
µXRF mapping is a useful tool for screening for diagenesis, but fails to pick up the fine details that
are visualized by CL-microscopy. Similarly, Mn and Fe profiles in µXRF line scanning will miss or
average out the small layers of vesicular calcite present in some parts of the shell hinges of *P.*
*vesicularis* and CL-microscopy remains a necessary tool for thorough screening for diagenesis.
**6.1.2 Diagenesis in trace element profiles**





Quantitative XRF line scans through the hinge of the *P. vesicularis* specimens show that absolute
concentrations of Fe and Mn rarely exceed 800 μg/g in all shells except for M11 (**Figure 5**). While Mn
concentrations measured in the hinges of *P. vesicularis* are higher than is considered typical for well-
preserved bivalve calcite and often exceed the diagenesis threshold of 300 μg/g proposed by
Steuber (1999), high concentrations of Sr (>700 μg/g) and Mg (>1000 μg/g), comparatively low Fe
concentrations and the observation of non-luminescent, well-preserved foliated calcite crystals
(**Figure 3**) suggest preservation of the original trace element signature (Veizer, 1983; Al-Aasm and
Veizer, 1986; Steuber, 1999). The peaks of high Fe concentrations in the M11 shell and elevated Fe
concentrations in M6 compared to the other shells coincide with decreases in $\delta^{18}O$ and $\delta^{13}C$. In
general, stable isotope values are lower in intervals of the records characterized by elevated levels of
Mn and Fe that exceed the baseline variation. Similarly, concentrations of Sr are generally lower in
samples with higher Mn concentrations (**Figure 6B**). This trend is especially clear in samples of which
Mn concentrations exceed 800 μg/g. This suggests that in these specimens of *P. vesicularis*, Fe and
Mn concentrations exceeding 800 μg/g likely signify areas where recrystallization has occurred (see
also **Figure 6A-C**). We therefore propose 800 μg/g as a tentative maximum threshold for the
preservation of pristine calcite in shells of *P. vesicularis* in this setting, and consider samples
exceeding this threshold in concentration for either Mn or Fe as diagenetically altered. Except for a
few measurements in shells M6 and M11, low Si concentrations and high Ca concentrations in the
trace element records shown in **Figure 5** indicate limited incorporation of detrital material into the
hinge of the shell (see de Winter and Claeys, 2017; de Winter et al., 2017a). This shows that the
infills of bore holes by detrital material have not significantly influenced the chemical signal of the
hinges of the shells. Indeed, the locations of these bore holes away from the shell hinge are
observed in **Figure 2 and 3.** From this it follows that the majority of post-mortem alteration of the
shells occurred through the process of chemical alteration (e.g. recrystallization) rather than physical
processes (e.g. predatory burrowing). As described above (see 5.1.1), the role of bore holes in the
shells (especially M6 and M11) in the diagenetic process was predominantly to provide entries
through which pore waters could enter to cause recrystallization. Bore holes elsewhere in the shells
may lead to migration of fluids through the shell, ultimately resulting in elevated concentrations
throughout the shell.

### 6.1.3 Diagenesis in stable isotope records

The majority of the stable isotope ratios measured the shell records are in agreement with those of
well-preserved Low Magnesium Calcite (LMC) of fossil (Steuber, 1996; 1999; Tripati et al., 2001) and
modern marine mollusc shells (Klein et al., 1996a;b; Goodwin et al., 2001; Lécuyer et al., 2004). The
low $\delta^{18}O$ and $\delta^{13}C$ values characterizing the central part of the M6 shell hinge record is an exception
to this and these values are likely explained by incorporation of vesicular calcite into the micromilled
samples. It is evident from the scan image of M6 in **Figure 2** how an extension of this shell mineral
phase into the umbo has resulted in the sampling of vesicular calcite in the centre of the record. The
resulting sudden decrease in $\delta^{18}O$ and $\delta^{13}C$ towards values below -4‰ and 1‰ respectively (a drop
of 2-3‰ for $\delta^{18}O$ and 1-2‰ for $\delta^{13}C$) illustrates that stable isotope composition of this vesicular
calcite deviates significantly from that of the foliated calcite. Similarly, the record from specimen M4
also has several stable isotope samples that most likely contain vesicular calcite. Lobes of vesicular
calcite in this specimen extend close to the hinge line, making incorporation of this microstructure
more likely. Several samples in the isotopic record of M4 are indeed characterized by unusually low
isotopic values. We consider it likely that small amounts of vesicular calcite were incorporated in
these samples.



The exceedingly low δ¹⁸O values in some samples from the vesicular calcite suggests that the original
composition is either not preserved due to alteration or that this vesicular calcite was initially
precipitated in disequilibrium with respect to ambient sea water (Grossman and Ku, 1986; Woo et
al., 1993; Steuber, 1999). The latter could be in agreement with the hypothesis that vesicular
structures in oyster shells are formed by microbes instead of by the bivalve itself (Vermeij, 2014).
However, microscopic images of the vesicular structure reveal blocky calcite crystals in some areas
(**Figure 3D**), which suggest recrystallization (e.g. Folk and Land, 1975; Schlager and James, 1978).
Indeed, the offset in stable isotope ratios of vesicular calcite compared to foliated calcite is not
found in modern oyster shells (Surge and Lohmann, 2008; Ullmann et al., 2010), and is therefore
most likely a result of preferential diagenetic alteration of the vesicular calcite. Elevated Mn and Fe
concentrations found in XRF mapping (**Figure 3**), and the notion that similar chalky or vesicular
phases in modern oyster shells are less crystalline and grow faster (Chinzei and Seilacher, 1993;
Ullmann et al., 2010), further attest to the fact that vesicular calcite in *P. vesicularis* (and likely in
other fossil members of the Gryphaeidae) is more prone to diagenetic alteration than its foliated
counterpart, and therefore provides no suitable record of palaeoclimatic information.
This conclusion is also supported by the clumped isotope analysis results. Bulk samples from the
ventral margin of the shell (containing more vesicular calcite, see **Figure 2 and 3**) contain lower
stable isotope ratios and higher reconstructed temperatures than samples from the dense shell
hinge (**Table 1**;  **Figure 7**). Elevated temperatures in these samples likely reflect recrystallization of
shell material from slightly warmer pore fluids after burial. However, temperatures from diagenetic
samples (average = 23°C) are relatively low compared to typical pore fluid temperatures measured
from diagenetic calcite in other studies (30-120°C; Huntington et al., 2011; Loyd et al., 2012; Dale et
al., 2014). Together with the fact that the difference between altered and unaltered samples (23°C
vs. 11°C; **Table 1**) is relatively small and that the dense calcite portions seem to be unaffected by
diagenesis, which suggests that burial was shallow. The shallow burial history is also demonstrated
by the preservation of organic biomarkers in the Bajada de Jagüel section (Woelders et al., 2017).
**6.1.4 Implications for sampling strategy**
Contrary to what may be expected based on the XRF map of M11 in **Figure 3**, the incorporation of
vesicular calcite into the microdrilled samples of M6 is not always reflected in elevated Mn and Fe
concentrations in the μXRF line scans. This could suggest that trace element signatures in vesicular
calcite this close to the shell hinge are not strongly affected by the leaching of reducing pore waters
that likely elevated the concentrations of these elements in the vesicular calcite of the rest of the
shell. Alternatively, it is likely that more of the vesicular calcite was incorporated in the microdrilled
samples for stable isotopes, than in the XRF line scan, as the line scan is only 25 μm wide and
relatively close to the hinge line, whereas the linear sampling paths of the microdrilling covered a
much larger area (up to 2 mm wide parallel to the growth increments). The wide sampling line
needed to sample for stable isotope analysis at this spatial resolution (100 μm in the direction of
growth) increases the chance of incorporating vesicular calcite into the samples, particularly in
samples further away from the hinge line and in shells where vesicular calcite layers penetrate close
to the hinge line (e.g. M4 and M11, see **Figure 2, Figure 3 and Figure 6D**). This result illustrates a
disadvantage of the abrasion-style microdrilling method applied in this study for spatially
heterogeneous bivalves. It shows that thorough screening for diagenesis using both trace element
analysis and cathodoluminescence is essential to correctly interpret the stable isotope results.
Summarizing, shells M6 and M11 are characterized by elevated Fe and Mn concentrations in the
shell hinge line, signifying that these specimens contain larger amounts of recrystallized vesicular
calcite in their shell hinge. Specimen M4 shows lower Fe and Mn concentrations in the shell hinge,



but low stable isotope ratios show that several microdrilled samples contain diagenetically altered
vesicular calcite. Stable carbon and oxygen isotope ratios in shells M4, M6 and M11 all show a
significant positive relationship, while such a relationship is absent in M0. As a result, of the 4
specimens investigated, specimen M0 is considered to represent the best preserved specimen, most
likely providing the most reliable results in terms of palaeoenvironmental reconstruction. Coloured
vertical bars in **Figure 5** illustrate parts of the shell records that were considered diagenetically
altered based on one or more of the criteria described above: 1) Bright luminescence in CL-
microscopy. 2) elevated (>800 µg/g) Fe and/or Mn concentrations. 3) Elevated Si (>0.5 mass%) and
reduced Ca (<38 mass%) concentrations. 4) Decreased stable isotope ratios ($\delta^{18}O$ < -3‰ and $\delta^{13}C$ <
1.5‰).
**6.2 Periodic variations**
**6.2.1 Shell chronology**
While earlier studies have been successful in determining the chronology of geochemical records
from comparatively young (Quaternary) fossil bivalves (e.g. Scourse et al., 2006; Marali and Schöne,
2014), attempts at palaeoseasonality reconstruction based on more ancient (pre-Quaternary) shells
have shown that this is not straightforward (Dettmann and Lohmann, 1993; Bougeois et al., 2014; de
Winter and Claeys, 2016; de Winter et al., 2017a). Quasi-periodic variations in stable oxygen
isotopes, Sr/Ca ratios and Mg/Ca ratios seem to represent seasonal cycles in shell growth (**Figure 5**),
but on closer inspection it is difficult to find a consistent phase relationships between these records
through all four shells. The most well-preserved shell record (M0) was tentatively subdivided into
annual cycles based on Sr/Ca and $\delta^{18}O$ seasonality. **Figure 8** shows a stack of the trace element
records created based on these subdivisions. Similar year stacks of the other three shells yielded
different phase relationships between proxies (**supplementary material**). These differences are
likely explained by the incorporation of diagenetically altered vesicular calcite in some of the
microdrilled samples, resulting in significantly lighter carbon and oxygen isotopic values. Especially in
the record of shell M4 (**Figure 5**), it is clear how diagenesis can preferentially influence one season
over the other and result in a change of the phase relationship between proxies in the shell. In the
case of M4, the incorporation of lobes of vesicular calcite into the shell hinge seems to be paced to
the seasonal cycle, making it difficult to disentangle patterns in diagenetic alteration from seasonal
patterns in the shell records. The incorporation of diagenetically altered vesicular calcite into the in
the shell hinge has influenced stable isotope profiles in shells M4, M6 and M11 more than M0, as is
evident from the significant correlation between $\delta^{18}O$ and $\delta^{13}C$ in these shells, which is absent in M0
(**Figure 6D**). Such preferential incorporation of vesicular calcite into the hinge during one season can
occur when the bivalve experiences more physiological stress in that season (Müller, 1970). Indeed,
even when diagenetically altered parts of these records (according to the threshold of 800 µg/g for
Fe and Mn and -3‰ for $\delta^{18}O$) are excluded, seasonal patterns in year stacks of shells M4, M6 and
M11 do not fully agree with those in the better preserved M0 shell. This leads to the assumption
that poorer preservation prevents the establishment of  a reliable chronology for these shells. That
said, records from shells M4, M6 and M11 should not be dismissed, as variation in the geochemical
proxies measured in pristine parts of these shells could still yield valuable information about the
extent of seasonality during their growth, even though phase relationships are blurred by diagenetic
overprinting. The fact that stable isotope measurements in these shells were not taken from the
exact same location as trace element measurements (due to different sampling and measurement
techniques) further complicates the establishment of consistent phase relationships between
geochemical records in the shells. The most obvious way in which this affected phase relationships
between records is the fact that stable isotope samples were more severely laterally averaged (2



mm wide transect compared to 25 µm wide transect of µXRF measurements), and the fact that
stable isotope records were rescaled to the length of XRF records before being plotted in **Figure 5**
(see section 4.5).
**6.2.2 Phase relationships**
Since only one of the shells measured in this study (M0) showed good enough preservation for a
discussion of phase relationships between records, care must be taken in extrapolating the
conclusions drawn from the year stack of this single shell. However, a tentative discussion of these
phase relationships may still shed some light on the mechanisms that drive the incorporation of
these proxies into the shell of *P. vesicularis*. The year stack of the well-preserved specimen M0
(**Figure 8**) shows that the $\delta^{18}$O, $\delta^{13}$C and Sr/Ca records exhibit a sinusoidal pattern with one peak per
year. In contrast, records of Zn/Ca, S/Ca and Mg/Ca contain a double peak in each year. Comparing
these observations with the records in **Figure 5** shows that the same seems to be true for the
pristine parts of the other three shells. In addition, the M0 year stack shows that maxima in $\delta^{13}$C
ratios coincide with minima in Sr/Ca and Zn/Ca and that minima in $\delta^{18}$O ratios follow maxima in $\delta^{13}$C
after about one quarter of an annual cycle. Zn/Ca and S/Ca records show an antiphase relationship,
and the Mg/Ca record has one minimum that coincides with a minimum in $\delta^{18}$O ratios and another
half a cycle earlier.
**6.3 Interpreting geochemical records in *Pycnodonte vesicularis***
6.3.1 Comparison with other taxa
Carbon isotope values found in this study are higher than in oysters living in modern coastal
temperate environments (Surge et al., 2001; Ullmann et al., 2010), but more similar to oysters living
in warmer, high-salinity or tropical settings (Klein et al., 1996a; Surge and Lohmann, 2008; Titschack
et al., 2010). Oxygen isotope ratios are generally lower than modern coastal mid latitude bivalves
(Ullmann et al., 2010; Klein et al., 1996b) and in better agreement with warmer, low latitude studies
(Lécuyer et al., 2004) and other Cretaceous bivalves (Steuber, 1999). This is in agreement with
reconstructions of $\delta^{18}$O ratios in Late Cretaceous oceans that were ~1‰ lower compared to the
present-day ocean due to the absence of extensive polar ice sheets in the Late Cretaceous (e.g. Hay,
2008). These results are in agreement with the warmer palaeoenvironmental setting inferred for the
Late Cretaceous of Neuquén Basin, based on TEX$_{86}$-palaeothermometry (Woelders et al., 2017).
However, the clumped isotope thermometry results of this study suggests rather cooler
temperatures. In order to properly interpret geochemical records from *P. vesicularis*, it is important
to compare the results of this study with those from closely related bivalves. Although the genus
*Pycnodonte* has no living members, two sister taxa in the subfamily Pycnodonteinae (Stenzel, 1959)
contain extant members: *Hyotissa* and *Neopycnodonte* (Stenzel, 1971).
6.3.2 *Hytissa hyotis*
The microstructure of *Hyotissa hyotis* is similar to that of *P. vesicularis*, with porous vesicular phases
alternated with dense foliated calcite layers. A specimen of *Hyotissa hyotis* in the northern Red Sea
was subject to a stable isotope study by Titschack et al. (2010). That study illustrates that, in contrast
to what was argued by Nestler (1965), the microstructure alternations in pycnodontein bivalves do
not correlate to annual growth increments. In the specimen of *H. hyotis* (Titschack et al., 2010),
seasonal variations in $\delta^{18}$O and $\delta^{13}$C were found to be independent of shell microstructure. Similarly,
in modern oysters like *Crassostrea virginica* (Surge and Lohmann, 2008) and *Crassostrea gigas*
(Ullmann et al., 2010), no isotopic difference is observed between different shell microstructures
(foliated vs vesicular calcite). This may explain why year stacks of *P. vesicularis* shells that were



affected by diagenesis differ from those of the well-preserved M0 specimen. The isotopically lighter
values observed in the vesicular calcite of *P. vesicularis* result from recrystallization, not of annual
cyclicity, and this incorporation of diagenetically altered samples into the record disturbed the
original stable isotope seasonality. Stable carbon isotope ratios in *H. hyotis* are very similar to those
measured in *P. vesicularis*. In principle, the $\delta^{13}$C signal of shells is controlled by the $\delta^{13}$C value of the
dissolved inorganic carbon (DIC) of the organism's extrapallial Fluid (EPF), from which the shell is
precipitated (Kirby, 2000). In marine bivalves, the carbon isotope composition of the EPF is
controlled by the $\delta^{13}$C of ambient seawater, carbonate ion effects, pH, food availability, growth,
valve gape/closure intervals, and seasonal changes in metabolic rate (Romanek et al. 1992;
McConnaughey et al. 1997; Kirby et al. 1998; Owen et al. 2002; Geist et al. 2005; McConnaughey and
Gillikin, 2008; Lartaud et al. 2010b). All these processes vary in strength and time, which complicates
interpretation of the $\delta^{13}$C signal (Lorrain et al. 2004; Omata et al. 2005). According to Titschack et al.
(2010), $\delta^{13}$C$_{shell}$ of *H. hyotis* is most strongly controlled by bivalve respiration, which is increased
during periods of enhanced planktonic food supply. They recorded a shifted phase relationship
between $\delta^{18}$O and $\delta^{13}$C records in *H. hyotis* similar to the phase shift observed in **Figure 8**, which was
attributed to phase-shifted cycles in sea surface temperature and productivity. In *H. hyotis* there is
an anti-phase relationship between $\delta^{13}$C and daily sunshine hours, suggesting that in our records of
the closely related *P. vesicularis*, the lowest annual $\delta^{13}$C values likely correspond to mid-summer
(December).
6.3.3 *Neopycnodonte zibrowii*
A specimen of *Neopycnodonte zibrowii* (Videt, 2004) was subject to detailed multi-proxy analysis in
Wisshak et al. (2008). This large, deep dwelling (450–500m) bivalve from the Azores shows similar
alternations in vesicular and foliated calcite as *P. vesicularis*, but has a much longer lifespan. Trace
element records in *N. zibrowii* show much higher Mg/Ca and S/Ca and lower Sr/Ca ratios compared
to those found in *P. vesicularis* in this study. Consecutive peaks in Mg/Ca and S/Ca coinciding with
minima in Ca and Sr concentration in this shell are interpreted as a sign of a strong control of growth
and reproductive cycle on trace element ratios. The covariation of Mg/Ca and S/Ca records in bivalve
calcite has often been interpreted as evidence of internal control on trace element concentrations
rather than external forcing (e.g. by temperature; Lorens and Bender, 1980; Rosenberg and Hughes,
1991). Such relationships between Mg/Ca, S/Ca and Sr/Ca are, however, not observed in *P.*
*vesicularis* (**Figure 5**, **Figure 6** and **Figure 7**). While $\delta^{13}$C values in *N. zibrowii* are similar to those
found in this study, $\delta^{18}$O in *N. zibrowii* are much higher and are interpreted to be controlled by
strong vital effects (Wisshak et al., 2008). Contrary to other modern oyster studies (Surge and
Lohmann, 2008; Ullmann et al., 2010; Titschack et al., 2010), Wisshak et al. (2008) do report an
isotopic offset between vesicular and foliated calcite, but $\delta^{18}$O values in vesicular calcite are
reported higher than in foliated calcite, opposite to what was observed in the specimens in this
study (**Figure 5**). A strong ontogenetic trend in $\delta^{13}$C observed in the juvenile part of *N. zibrowii*
records is again opposite to the trend in $\delta^{13}$C observed in this study. This shows that the common
explanation of incorporation of isotopically light $CO_2$ into the shell due to enhanced metabolic rate in
the juvenile stage (e.g. Jones et al., 1986; Lorrain et al., 2005; Gillikin et al., 2007; Wisschak et al.,
2008) does not explain the $\delta^{13}$C trend in M6 and M11 shells in this study (**Figure 5**). Instead, any
trends in $\delta^{13}$C in these shells are most likely caused by the effects of sampling and incorporating
recrystallized vesicular calcite into the stable isotope samples, which is also evident from the
elevated Fe concentrations in these shells. The fact that Fe concentrations in M11 are highest in the
ontogenetically oldest part of the record further confirms that the observed drop in stable isotope
values towards the ontogenetically oldest part of this record is caused by diagenesis, and is not an
ontogenetic trend. This is in agreement with work on extant oysters, in which such an ontogenetic



trend is generally absent (Surge et al., 2001; Surge and Lohmann, 2008; Ullmann et al., 2010). The
vast difference in geochemical records between these closely related bivalve taxa shows that vital
effects (growth and metabolic rates) play a large role in their mineralization, and that independent
control on the growth rates of these bivalves could be crucial in disentangling internal from external
forcing in bivalve shells. In terms of their expression of proxy records and their environmental niche,
records from *P. vesicularis* shells obtained in this study show much closer resemblance to those of *H.*
*hyotis* and marine *Crassostrea gigas* (Surge and Lohmann, 2008; Ullmann et al., 2010) than to those
of *N. zibrowii*, making *H. hyotis* the best modern analogue to compare with records from shallow
marine *Pycnodonte* shells.
6.3.4 Timing of shell deposition and seasonality
The $\delta^{18}O$ values of the specimens of *H. hyotis* studied by Titschak et al. (2010) are higher than the
$\delta^{18}O$ values of our specimens of *P. vesicularis*. Presumably, this is because the specimens studied by
Titschak et al. (2010) grew in an environment characterized by a strong evaporatic setting (Safaga
Bay). This setting likely resulted in a higher salinity and $\delta^{18}O_{seawater}$ (2.17‰) than in the Neuquén
Basin (-2.8‰ based on clumped isotope results from well-preserved shells in this study). As a
consequence, $\delta^{18}O$ records *H. hyotis* in Titschack et al., (2010) are strongly correlated with both Sea
Surface Temperature (SST) and Sea Surface Salinity (SSS). Such an interplay of salinity and
temperature on stable isotope composition of bivalve calcite has been studied in *Crassostrea*
*virginica* that grew under changing salinity conditions (Surge et al., 2001). However, in contrast to
estuarine *C. virginica* studied by Surge et al. (2001), where both stable isotope records are in phase,
the best preserved specimen in our study (M0) presents a (shifted) anti-phase relationship between
$\delta^{18}O$ and $\delta^{13}C$. Following the rationale that annual lows in $\delta^{13}C$ occur in mid-summer in *P. vesicularis*,
this would suggest that the lowest $\delta^{18}O$ values are reached in spring (September-November). As $\delta^{18}O$
is negatively correlated to temperature and positively correlated to salinity, this would suggest that
$\delta^{18}O_{shell}$ variations in our records are more strongly forced by changes in salinity rather than in
temperature, since sea surface temperature is unlikely to be higher in spring than in summer. If so,
our record suggests that the Neuquén basin experienced a decrease in salinity in the spring. Highest
salinities are reached in summer and autumn and lowest salinities in winter to spring, possibly
corresponding to a winter-spring precipitation maximum similar to the present day situation at this
latitude in this region (Servicio Meteorológico Nacional, 2017).
6.3.5 Palaeoproductivity
The fact that a minimum in Zn/Ca coincides with a maximum in S/Ca and $\delta^{13}C$ and a minimum in $\delta^{18}O$
in the well preserved M0 specimen (**Figure 5 and 6**) is in agreement with the proposed explanation
of these seasonal records. Zn concentrations in bivalve shells drop during a productivity bloom,
which occurs in the spring season (September-November; Calvert and Pedersen, 1993; Jackson et al.,
1993; Guo et al., 1997, de Winter et al., 2017a). The observation that a minimum in Zn/Ca coincides
with the lowest $\delta^{18}O$ values which occurred in spring and precedes the minimum in $\delta^{13}C$ that
occurred in mid-summer is consistent with the hypothesis of spring blooms affecting the amount of
bio-available Zn in the surface ocean and forcing Zn/Ca ratios in the shells of *P. vesicularis* (Guo et
al., 2002). This explanation is further supported by the timing of the onset of the drop in Zn/Ca
synchronous with a maximum in $\delta^{13}C$. The annual $\delta^{13}C$ cycle in the closely related *H. hyotis* was also
proposed to reflect a seasonality in productivity by Titschack et al. (2010), showing that the drop in
Zn/Ca may indeed be related to a spring bloom in productivity. Increased fresh water input into the
basin during spring, which caused the low salinity conditions that are observed in the $\delta^{18}O$ records,
could have provided the nutrients that initiated this productivity bloom. Seasonal decreases in
salinity are in agreement with reconstructions by Woelders et al. (2017).



6.3.6 Physiological effects
The observed anticorrelation between $\delta^{18}O$ and S/Ca in specimen M0 suggests that in *P. vesicularis*,
S/Ca responds as a physiological parameter that co-varies with seasonal changes, such as food
availability, growth or respiration rate. This response has also been inferred for other groups of
bivalves, where S/Ca ratios were considered to reflect metabolic rates (e.g. Rosenberg and Hughes,
1991). A peak in S/Ca during the spring season, when a productivity bloom coincides with a potential
decrease in salinity is in agreement with this explanation. Such environmental perturbations affected
the growth of the bivalve and have been linked to an increase in the incorporation of sulphur into
the organic matrix of the bivalve shell (Lorens and Bender, 1980). The fact that the amplitude of S/Ca
variations in the record of M6 increases in the part of the shell where vesicular calcite penetrates
the shell hinge (**Figure 5**) supports the hypothesis that these disturbances of the shell hinge indicate
periods of physiological stress experienced by the bivalve (Müller, 1970). Interestingly, the year stack
of specimen M0 shows a smaller second peak in Zn/Ca and S/Ca that coincides with autumn if the
interpretation of phase relationships between records is correct. This may reflect a smaller
productivity bloom in autumn. Similarly, a decrease in Sr/Ca ratios synchronous with the peak in $\delta^{13}C$
suggests a physiological origin of the seasonality in this proxy. The fact that Sr/Ca ratios are lower
during the low-salinity, high-productivity spring season in which growth was probably slower is in
agreement with relationships between Sr/Ca and growth rate found in modern bivalves (e.g. Lorrain
et al., 2005; Gillikin et al., 2005a). As mentioned above, care must be taken to extrapolate these
interpretations since they are based on only one well-preserved shell.
**6.4 Temperature proxies**
An overview of all temperature proxies used in this study is plotted in **Figure 9**. This figure illustrates
some of the complexity of combining these different proxies in *P. vesicularis* to reconstruct
seasonality. Combination of the $\delta^{18}O_{sw}$ values reconstructed using clumped isotope analysis with the
high-resolution $\delta^{18}O$ records yields a tentative sub-annual palaeotemperature reconstruction for all
shell records. However, the variations in these records may not reflect true sub-annual temperature
variations, especially since it is likely that salinity in the Neuquén Basin did not remain constant
through the year (see 5.3.4). Temperature reconstructions based on clumped isotope and $\delta^{18}O$
records are systematically lower than the $TEX_{86}{}^H$ temperatures. This offset can partially be explained
by the fact that $TEX_{86}{}^H$ is calibrated to sea surface temperatures while *P. vesicularis* lived 50-75 m
below sea level (Scasso et al., 2005), in waters that were likely slightly cooler than those at the sea
surface. However, this difference is most likely not enough to explain the offset of ±15°C between
clumped isotope and $TEX_{86}{}^H$ temperature reconstructions. Over the past years, several studies have
highlighted the complexity of shallow marine $TEX_{86}$ records and have shown that temperature
reconstructions by this method may be biased (e.g. Jia et al. 2017). Similarly, in the compilation
study of O'Brien et al. (2017), Cretaceous $TEX_{86}$-based sea surface temperatures are systematically
higher than planktic foraminiferal $d^{18}O$-based temperatures. In some settings, $TEX_{86}$ has been shown
to predominantly reflect summer temperatures (Schouten et al., 2013). It is possible that also in the
Neuqúen basin $TEX_{86}{}^H$ reconstructed temperatures are biased towards summer-season
temperatures. In contrast, clumped isotope thermometry on our *P. vesicularis* specimens
reconstructs a mean value of the entire growth season of the bivalve. Yet, it is likely that growth in
these bivalves slowed or ceased during the spring and summer season (as is evident from Sr/Ca
ratios, see 5.3.6). The year stack in **Figure 8** also shows that low $\delta^{18}O$ values make up a much smaller
portion of the year than the higher $\delta^{18}O$ values, suggesting a growth stop in the low-$\delta^{18}O$ season. It is
therefore likely that temperature reconstructions of both clumped isotope thermometry and $TEX_{86}{}^H$
measurements are seasonally biased and that the mean annual temperature lies in between these



two estimates. Alternatively, more vesicular calcite might have been incorporated into the shell
hinge as a result of more stressful growth conditions (Müller, 1970; see 5.2.1) causing these warm
seasons to be selectively overprinted by diagenesis. Indeed, this seems to be the case in the record
of M4, where low values in $\delta^{18}O$, associated with the spring season, are more characterized by
diagenetic alteration than parts of the year (**Figure 9**). If vesicular calcite is avoided in clumped
isotope sampling, this will cause a bias towards colder seasons for clumped isotope temperatures.
However, in practice it will be difficult to avoid these lobes of vesicular calcite and small amounts are
likely to be included in clumped isotope samples, leading to higher palaeotemperature
reconstructions.
As mentioned above, several temperature calibrations exist for Mg/Ca ratios in bivalves. The most
likely candidates for temperature reconstruction based on Mg/Ca of *P. vesicularis* are the
calibrations based on ostreid bivalves. Promising examples are the calibrations by Mouchi et al.
(2013, based on juvenile specimens of the pacific oyster *Crassostrea gigas*) and Surge and Lohmann
(2008; based on *Crassostrea virginica*). A factor that complicates the interpretation of Mg/Ca ratios
in terms of temperature is the fact that sea water Mg/Ca (Mg/Ca$_{ocean}$) has changed over geological
timescales, and is thought to have been much lower in the late Maastrichtian than in the present-
day ocean (Maastrichtian Mg/Ca$_{ocean}$ of 1-2 mol/mol compared to 5 mol/mol in the modern ocean;
Stanley and Hardie, 1998; Coggon et al., 2010). This complicates the use of modern transfer
functions which were established for bivalves growing in modern ocean conditions. Since these
changing ocean compositions have most likely influenced Mg/Ca ratios in calcifying organisms (Lear
et al., 2015), temperature calibrations need to be corrected accordingly (de Winter et al., 2017a).
Therefore, here, Mg/Ca$_{ocean}$ ratios of ~1.5 mol/mol were used to represent average Maastrichtian
ocean water, about 3.3 times lower than in the modern ocean. With this correction, the *C. virginica*
temperature calibration by Surge and Lohmann (2008; **Figure 9**) approach reconstructions based on
the other proxies in terms of temperature seasonality, while the calibration of Mouchi et al. (2013)
seems to significantly overestimate temperature (MAT >60°C). Reconstructions based on the Mg/Ca
calibration of Surge and Lohmann (2008) yield sea water temperatures (20°C ± 10°C) slightly higher
than those observed in the $\delta^{18}O_{sw}$-corrected $\delta^{18}O$ record.
Since Mg/Ca ratios yield temperatures between clumped isotope and TEX$_{86}^{H}$ reconstructions, it is
tentative to assume that they more closely represent mean annual temperatures than the other
proxies. However, there are large differences (>10°C) between temperature reconstructions of
Mg/Ca and $\delta^{18}O$ in some parts of the records. Furthermore, the well-preserved M0 shell record
shows an anticorrelation between the seasonal fluctuations of the two temperature records in parts
of the record, suggesting that at least one of the proxies may largely be controlled by a factor other
than ambient temperature (although phasing arguments may be affected by the relative scaling of
trace element and stable isotope records). Seasonal changes in salinity cannot account for this
difference between the proxies, as a seasonal increase in salinity of approximately 20 PSU would be
required to account for the offset in temperature between the proxies (Ravelo and Hillaire-Marcel,
2007). Such a severe change in salinity is not consistent with earlier palaeoenvironmental
reconstructions in the Neuquén Basin (Prámparo et al. 1996; Prámparo and Papú 2006; Woelders et
al. 2017). Additionally, there seems to be no a priori reason why Mg/Ca temperature calibration of
Surge and Lohmann (2008) would be the most suitable calibration for *P. vesicularis*, which may
require its own species-specific calibration. If seasonal growth cessations are present in *P.
vesicularis*, they would affect Mg/Ca as well as $\delta^{18}O$ and cause Mg/Ca records to have the same
seasonal bias. It must be noted that the fact that trace element records and stable isotope records
were measured using different methods makes it possible that the records are slightly shifted with
respect to each other (see section 4.5). As a consequence, phase relationships between Mg/Ca and



δ[18]O temperature reconstructions may have been distorted. Closer observation of **Figure 9** indeed
shows that temperature reconstructions based on these two records are in some cases shifted with
respect to each other. This might explain part of the offset between the reconstructions and render
Mg/Ca temperatures more probable. Nevertheless, the uncertainties of Mg/Ca temperature
reconstructions in bivalves, together with the observed lack of temperature dependence of Mg/Ca
ratios in the closely related *Pycnodonte zibrowii*, leads to the conclusion that temperature
reconstructions based on Mg/Ca ratios in *Pycnodonte* oysters are difficult.
In summary, δ[18]O values in the shells of *P. vesicularis* have been shown to vary with changes in
salinity in this setting. Temperatures reconstructed by clumped isotope thermometry from well-
preserved parts of different bivalve shells agree and seem to be the most reliable method for
temperature reconstruction. These clumped isotope temperature reconstructions are in agreement
with present-day average annual temperatures in the region (~10-15°C; Servicio Meteorológico
Nacional, 2017), while they are slightly below model and proxy-based SST reconstructions for the
Maastrichtian mid-latitudes (20-25°C; e.g. **Donnadieu et al., 2006; Brugger et al., 2017; O'Brien et**
**al., 2017**). Comparison of all palaeotemperature proxies in this study shows that TEX$_{86}^{H}$ temperature
reconstructions (27-30°C) likely overestimate MAT, while clumped isotope thermometry might
underestimate it. Mg/Ca temperature reconstructions show promising results (15-20°C), but depend
heavily on the calibration that is used and are therefore considered problematic. The best approach
to reconstruct palaeotemperature seasonality from *Pycnodonte* shells would be to microsample the
foliated calcite of the shells for clumped isotope analysis. This microsampling can be guided by
records of conventional stable isotope ratios and trace element concentrations to ensure the
sampling of material from different seasons. Via this approach, both seasonality in temperature and
salinity can be reconstructed from *Pycnodonte* shells, and the effects of salinity and temperature on
δ[18]O values can be disentangled.
**7. Conclusions and recommendations**
This study represents a first attempt to employ the shells of the honeycomb oyster *Pycnodonte*
*vesicularis* for the reconstruction of late Maastrichtian palaeoseasonality. The multi-proxy approach
applied in this work demonstrates the complexity of such attempts to reconstruct
palaeoenvironmental conditions. Yet, this approach also demonstrates the value of using a range of
different methods to characterize the preservation state and chemical composition of fossil bivalve
calcite. Based on the results presented in this work, several recommendations can be made for the
use of shells from *P. vesicularis* for the reconstruction of palaeoseasonality and palaeoenvironment.
Detailed analysis of shell structure and preservation shows that shells of *P. vesicularis*, like other
species of the Order Ostreoida, are characterized by two major micromorphologies of calcite, which
were referred to by Carriker et al. (1980b) as "chalky" and "foliated" calcite. In the case of *P.*
*vesicularis*, CT scanning shows that these "chalky" (vesicular) calcite layers are characterized by a
high degree of porosity (up to 65%) and are therefore very permeable for pore fluids (**Figure 4**). The
thin walls of the vesicular calcite structure provide a lot of surface contact between permeating pore
fluid and the calcite, making it prone to recrystallization (**Figure 3**). The recrystallization and the
precipitation of secondary carbonates in this porous micromorphology therefore renders the
vesicular calcite of pycnodontein bivalves poorly suitable for palaeoenvironmental reconstruction. In
addition, pore fluid can enter the shells of *P. vesicularis* post mortem through bore holes, for
example made by polychaete worms. Subsequently, recrystallization and precipitation of secondary
carbonates in equilibrium with these reducing pore fluids raises the concentrations of Mn and Fe
(see XRF mapping and CL images in **Figure 3**) and lowers stable isotope ratios. Hence, when selecting
specimens of *P. vesicularis* for palaeoseasonality reconstructions, specimens affected by boring





organisms are best avoided or treated with care. Micro-analytical techniques such as
cathodoluminescence microscopy, optical microscopy and μXRF mapping allows to avoid these
zones of recrystallization.
Foliated calcite layers in the shell hinge of *P. vesicularis* are less affected by these diagenetic
processes and stable isotope, clumped isotope and trace element compositions of these layers
suggest preservation of primary calcite, making it suitable for palaeoseasonality reconstruction.
However, care must be taken in sampling these parts of the shells of *P. vesicularis*, as lobes of
vesicular calcite can extend into the hinge of the shells. Such lobes of vesulicar calcite can be very
thin, and can be difficult to avoid while microsampling for stable isotope ratios. Incorporation of
vesicular calcite into stable isotope samples will significantly alter the measured stable isotope ratios
and influence the interpretation of palaeoseasonality. Clumped isotope analysis of samples
containing this vesicular calcite yield much higher temperatures than samples of foliated calcite,
suggesting diagenetic overprinting of the stable isotope signal. The multi-proxy approach in this
study allows the distinction of diagenetic parts in fossil bivalve shells and aids in the evasion of
diagenetically altered parts of the shells and the consideration of only well-preserved parts.
Future work on *P. vesicularis* shells, as well as other gryphaeid shells that contain multiple
microstructures, aiming at the reconstruction of palaeoseasonality over geological time scales
should benefit from the application of a multi-proxy approach that allows the interpretation of
seasonally changing environmental parameters. However, the establishment of a shell chronology
from these records can be difficult, as selective diagenetic overprinting, the occurrence of growth
cessations and the complexity of synchronizing proxy records from multiple methods can complicate
the interpretation of phase relationships between proxies. Multi-proxy analysis on one exceptionally
well-preserved specimen demonstrates how the timing of seasonal deposition of the shell could be
determined from the phase relationships between proxies. If applied correctly, this approach also
allows the separation of the effects of, for example, temperature and salinity on the stable isotope
ratios in the shells. However, it must be noted that extrapolation of results from one well-preserved
specimen means that the interpretation of phase relationships in this study must remain tentative.
Even though the establishment of shell chronology for less well-preserved samples is difficult, multi-
proxy records from well-preserved parts of these shells can still yield information about the sub-
annual variation of proxies in *P. vesicularis*. Comparison of these multi-proxy shell records with
contextual proxy reconstructions allows palaeoseasonality reconstructions to be placed in a larger
geological context and allows the discussion of different palaeotemperature proxies.
Records of uncontaminated foliated calcite in the hinge of well-preserved specimens of *P. vesicularis*
yield a mean annual sea water temperature in the late Maastrichtian Neuquén Basin of 11°C, which
is lower than reconstructions based on contextual TEX$_{86}^{H}$ palaeothermometry (±27°C). This
comparison suggests that the TEX$_{86}^{H}$ method overestimates mean annual temperatures in this
setting, possibly representing summer surface water temperatures. Clumped isotope thermometry
of bulk foliated calcite samples likely underestimates the annual mean because the warm spring and
early summer season is underrepresented in the shells due to slower growth or growth cessations. A
seasonality in δ$^{18}$O of about 1‰ is ascribed to a combination of decreased salinity by fresh water
input in the spring season and a moderate temperature seasonality, but the aforementioned
seasonal bias prevents capture of the full seasonal cycle in this record. Attempts to verify the
seasonality in SST by Mg/Ca ratios of shell calcite are complicated by uncertainties about vital effects
on the incorporation of Mg into the bivalve shell. After correction for lower sea water Mg/Ca ratios
in the Late Cretaceous, Mg/Ca temperatures calculated using the oyster-based calibration of Surge
and Lohmann (2008) fall between temperatures of clumped isotope palaeothermometry and those



of TEX$_{86}^{H}$ palaeothermometry and reveal a pattern similar to the δ$^{18}$O records. While it is tentative to
conclude that this record most closely reconstructs the temperature seasonality, the uncertainties
involved in bivalve Mg/Ca records precludes such a straightforward conclusion.
This multi-proxy work shows that, even using several independent palaeotemperature
reconstruction methods, the reconstruction of temperature seasonality from fossil bivalve calcite is
strongly complicated by the influence of other palaeoenvironmental parameters that affect the
chemistry of bivalve shells. Yet, the successful application of clumped isotope thermometry on fossil
bivalve calcite in this study indicates that temperature seasonality in fossil ostreid bivalves may be
constrained by the sequential analysis of foliated calcite samples using this method.

**Acknowledgements**
Niels J. de Winter is financed by a personal PhD fellowship from IWT Flanders (IWT700). This
research was partly financed by the FOD40 Chicxulub grant obtained by Philippe Claeys. Robert P.
Speijer is funded by the Research Foundation Flanders (FWO grant G.0B85.13). Johan Vellekoop is
also funded by a personal research grant from FWO (grant 12Z6618N). Thanks go to the Hercules
foundation Flanders for acquisition of XRF instrumentation (grant HERC1309) and VUB Strategic
Research Program for support of the AMGC research group. The authors thank Prof. Rudy Swennen
from the KU Leuven for analytical support. The author declares that there are no conflicts of
interest.

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

[FIGURE CAPTIONS]

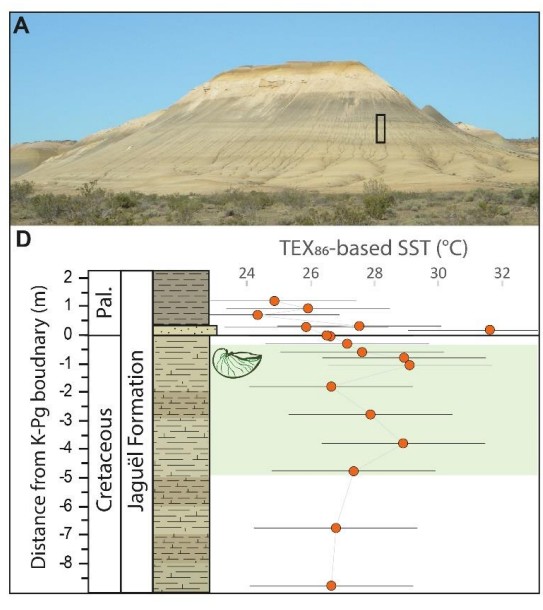

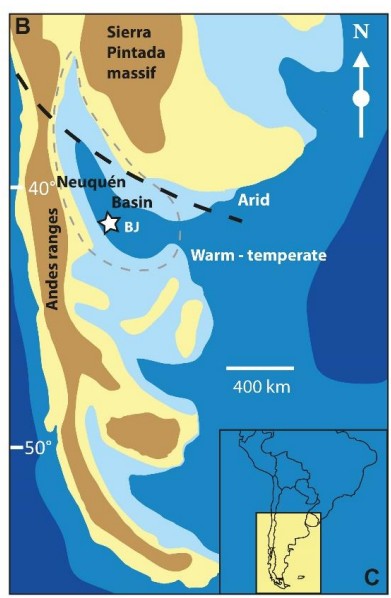

**Figure 1**
Origin and stratigraphy of the studied *Pycnodonte vesicularis* specimens. A) photograph of the
Bajada de Jaguël section (BJ; modern location: 38°06'10.5"S, 68°23'20.5"W, palaeolatitude =
43°S). B) Palaeogeography of study area during the latest Cretaceous. Palaeomap after
Scasso et al. (2005) and Woelders et al. (2017). C) Location of the study area in southern
Argentina relative to modern day South America. D) lithology, stratigraphy and TEX$_{86}$ record
(Woelders et al., 2017) of the BJ section. The main *P. vesicularis* level is indicated in light
green.



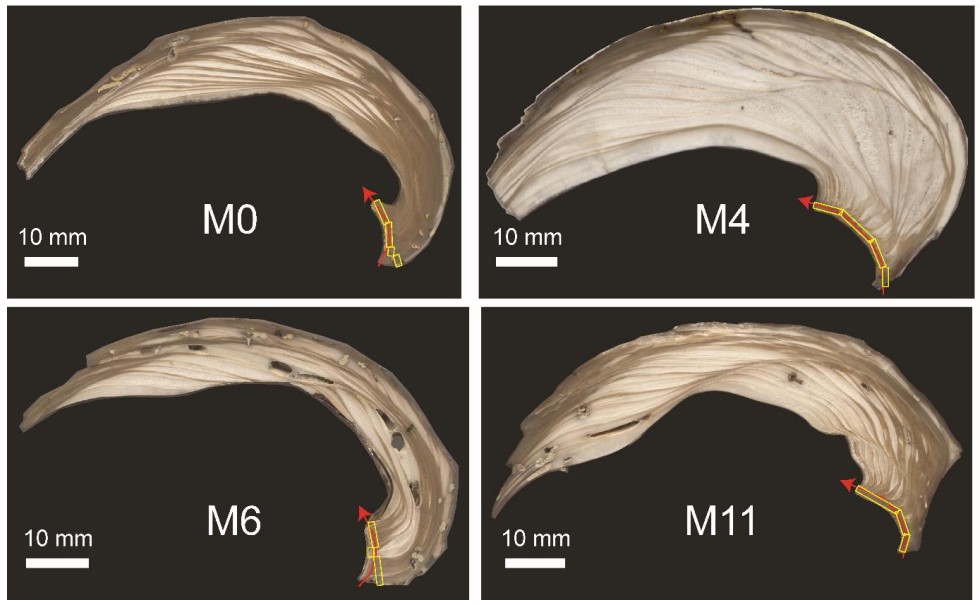

**Figure 2**

Colour scans of cross sections of the four shells subject to multi-proxy analysis. Red arrows indicate
sampling location and direction. Yellow boxes indicate the location of stable isotope
transects.



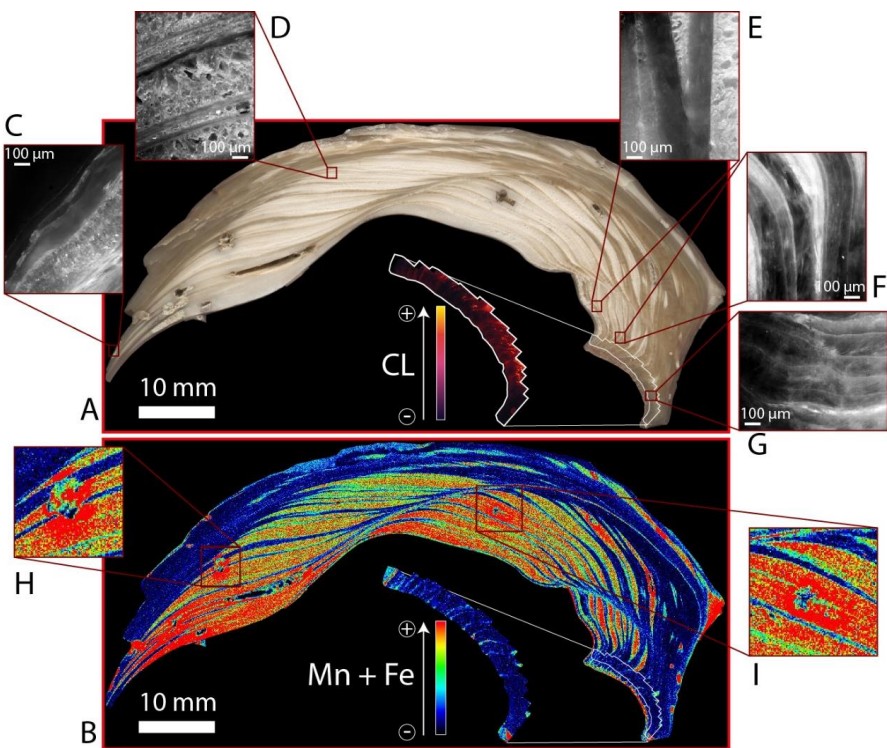

**Figure 3**

Overview of the results of colour scanning, microscopic analyses and µXRF mapping of specimen M11. A) Colour scan of cross section in growth direction through the shell, with close-up of cathodoluminescence microscopic image of the hinge line. B) µXRF mapping of the cross section, with close-up of the µXRF map of the hinge line. C) Optical microscopic image of transitions between dense foliated calcite and porous vesicular calcite. Note the blocky calcite crystals in the vesicular structure. D) Optical microscopic image of very thin, alternating layers of foliated and vesicular calcite. E) Optical microscopic image of sharp transitions between dense foliated calcite and porous vesicular calcite F) Optical microscopic image of more gradual transitions between foliated calcite and vesicular calcite. G) Optical microscopic image of dense, foliated calcite layers in shell hinge line. Note the thin layer of vesicular calcite (white) intercalated between the foliated layers near the bottom of the image. H) and I) Close-up of µXRF mapping of bore hole with corona of elevated Fe and Mn concentrations.





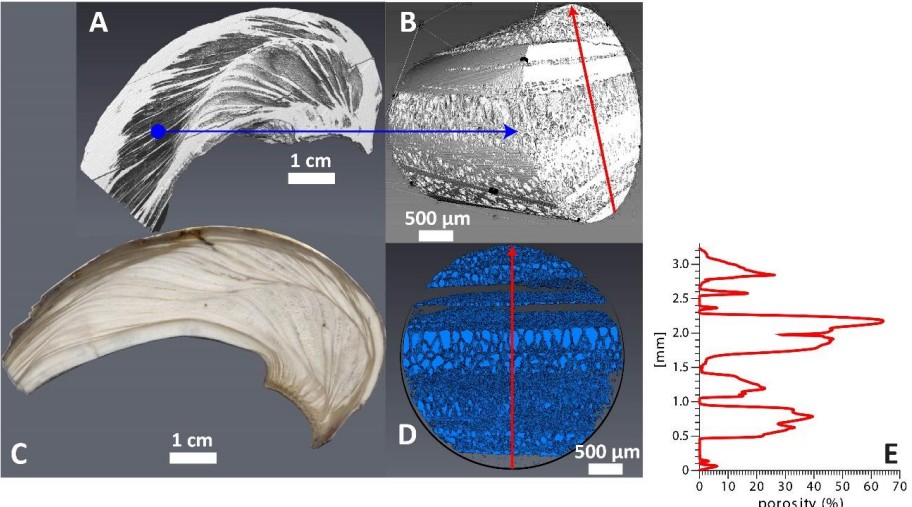

**Figure 4**

Overview of the results of CT-scanning and pore analysis on specimen M4, with A) showing an
overview of density variations in the shell (white = dense calcite, darker colours represent
porosity). The blue dot shows the location of the part of the shell that was CT-scanned at
high resolution. B) shows the shape and density of a part of the shell that was CT-scanned
with higher spatial resolution as well as the location of the porograph shown in E). C) shows
a colour scan of the shell cross section. D) shows a cross-section through the high-resolution
section through the shell with porosity in blue. E) shows a graph of porosity through the high
resolution section perpendicular to the growth layers.



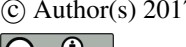

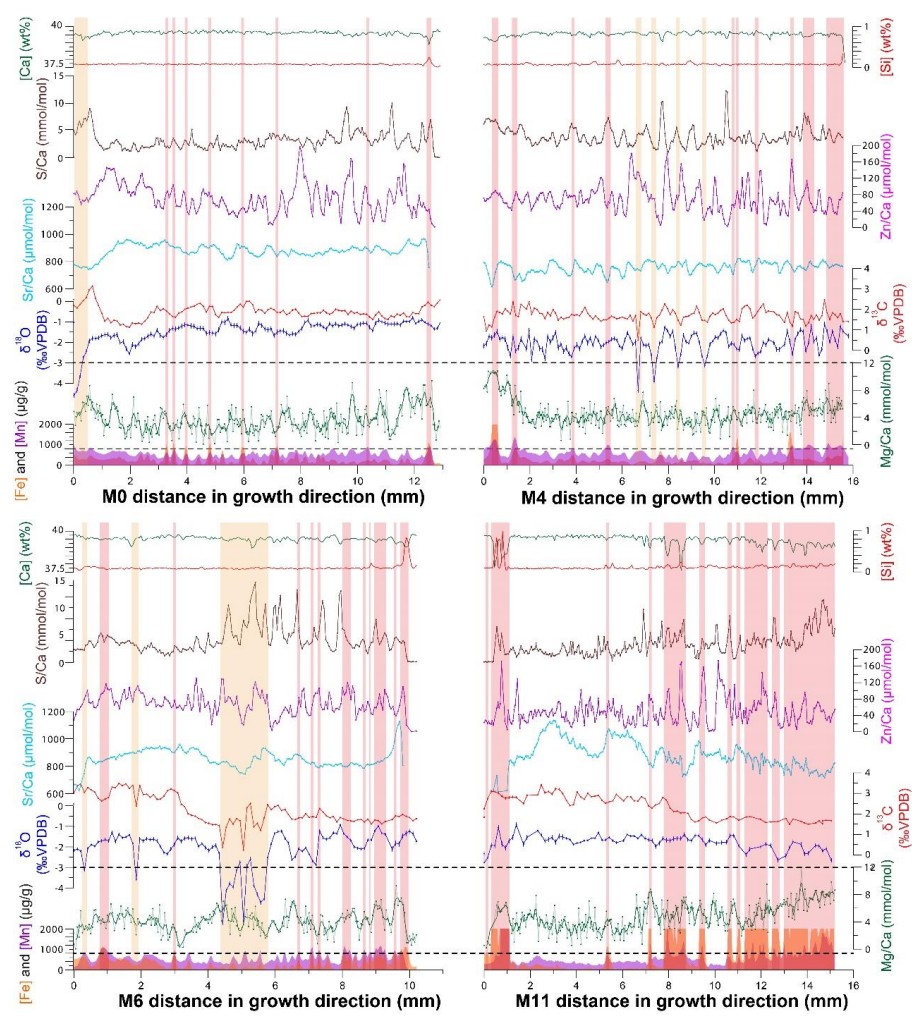

**Figure 5**

Overview of multi-proxy records through the hinges of 4 specimens of *P. vesicularis*. From top to
bottom, records of [Ca] (green), [Si] (red), S/Ca ratios (brown), Zn/Ca ratios (purple), Sr/Ca
ratios (light blue), $\delta^{13}C$ (red), $\delta^{18}O$ (blue), Mg/Ca (green), [Mn] (purple) and [Fe] (orange) are
shown. Red arrows in Figure 2 indicate the direction of sampling. Vertical bars indicate parts
of the records that were affected by diagenesis based on Mn and Fe concentrations (red
bars) and stable isotope ratios (orange bars). Note that the vertical scale of the Mn and Fe
plots is clipped at 2000 µg/g.



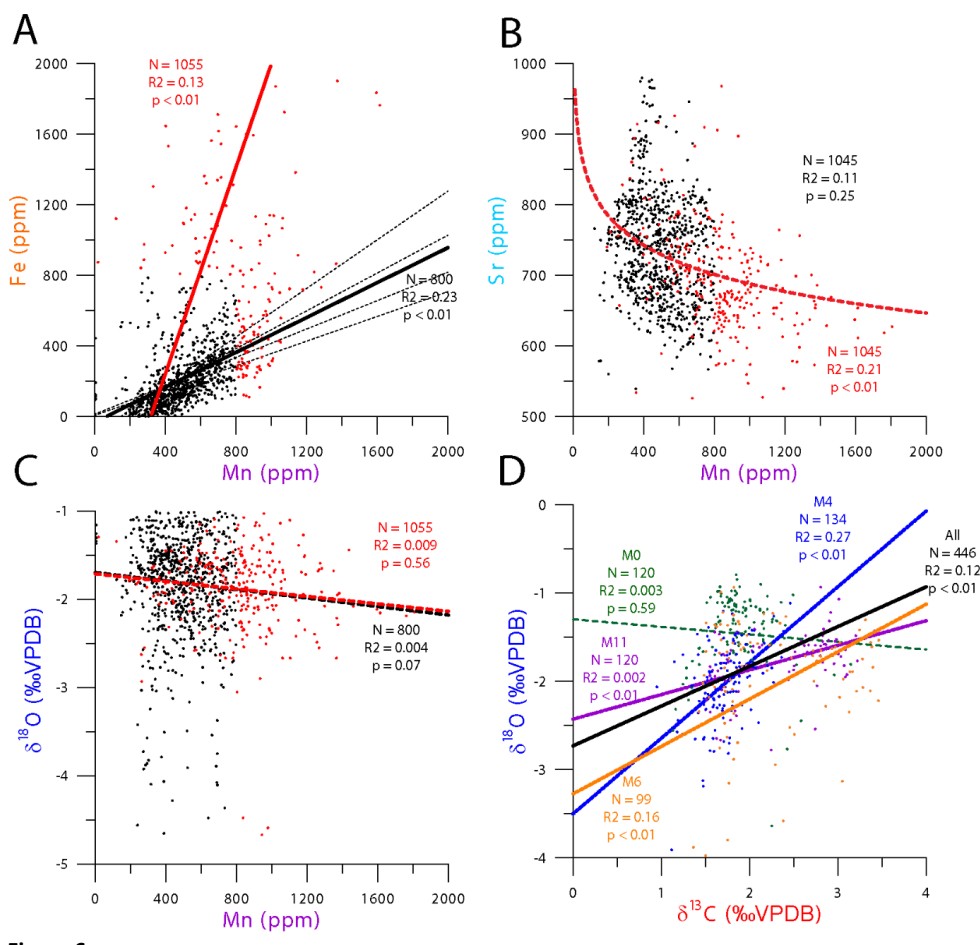

**Figure 6**
Cross plots showing cross plots between trace element and stable isotope measurements in the
shells. Black lines indicate correlations through all measurements, red lines show
correlations of diagenetically altered samples (according to the 800 µg/g threshold for Fe
and Mn) and alternatively coloured lines indicate correlations in individual shells. Statistics
of the regressions are indicated in matching colours. A) [Fe] vs [Mn] showing how both
elements increase with increasing diagenetic overprinting. Steeper slopes suggest relatively
more Fe is added in diagenetically altered samples. B) [Sr] vs [Mn] showing decreasing Sr
concentrations corresponding to increasing [Mn], but only in diagenetically altered samples.
C) $\delta^{18}O$ vs [Mn] showing lack of correlation, although Mn-rich diagenetic samples generally
have lower $\delta^{18}O$ values. D) $\delta^{18}O$ vs $\delta^{13}C$, showing positive correlation in specimens affected
by diagenesis and no correlation in M0, which has pristine values.





**Figure 7**
Cross plots of clumped isotope results. A) $\Delta_{47}$ vs. $\delta^{18}O$ from clumped isotope measurements on all
seven shells. Red dots and error bars represent measurements of samples from the ventral
margin of the shells, while black dots and error bars indicate results from dense foliated
calcite from the hinge of the shells. Dashed lines illustrate the $\delta^{18}O$ values of seawater that
correspond to the combination of $\Delta_{47}$ and $\delta^{18}O$ values in the graph. B) $\delta^{13}C$ vs. $\delta^{18}O$ from
clumped isotope measurements on all shells. Red dots and error bars represent
measurements of samples from the ventral margin of the shells, while black dots and error
bars indicate results from dense foliated calcite from the hinge of the shells. Numbers next
to the dots indicate $\Delta_{47}$ values measured in the same samples. Coloured rectangles indicate
the range of pristine stable isotope values measured in high resolution transects through the
hinges of shells M0, M4, M6 and M11.

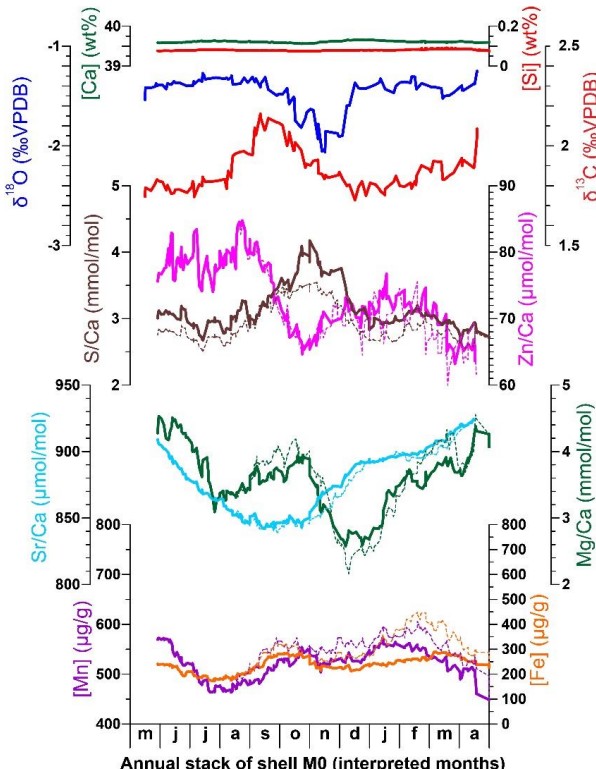

**Figure 8**
Stack of proxy records shell M0 made according to a tentative interpretation of annual cyclicity
based on Sr/Ca ratios in **Figure 5**. Solid lines indicate annual stacks excluding diagenetically
altered samples while dashed lines include all measured samples to show the effect of
diagenesis. From top to bottom, stacks of [Ca] (green), [Si] (red), $\delta^{13}C$ (red), $\delta^{18}O$ (blue), S/Ca
ratios (brown), Zn/Ca ratios (purple), Sr/Ca ratios (light blue), , Mg/Ca (green), [Mn] (purple)
and [Fe] (orange) records are shown. Subdivisions of the stack into 12 time steps and
corresponding months are based on an interpretation of the phase relationship between the
proxies in terms of palaeoenvironmental seasonality.



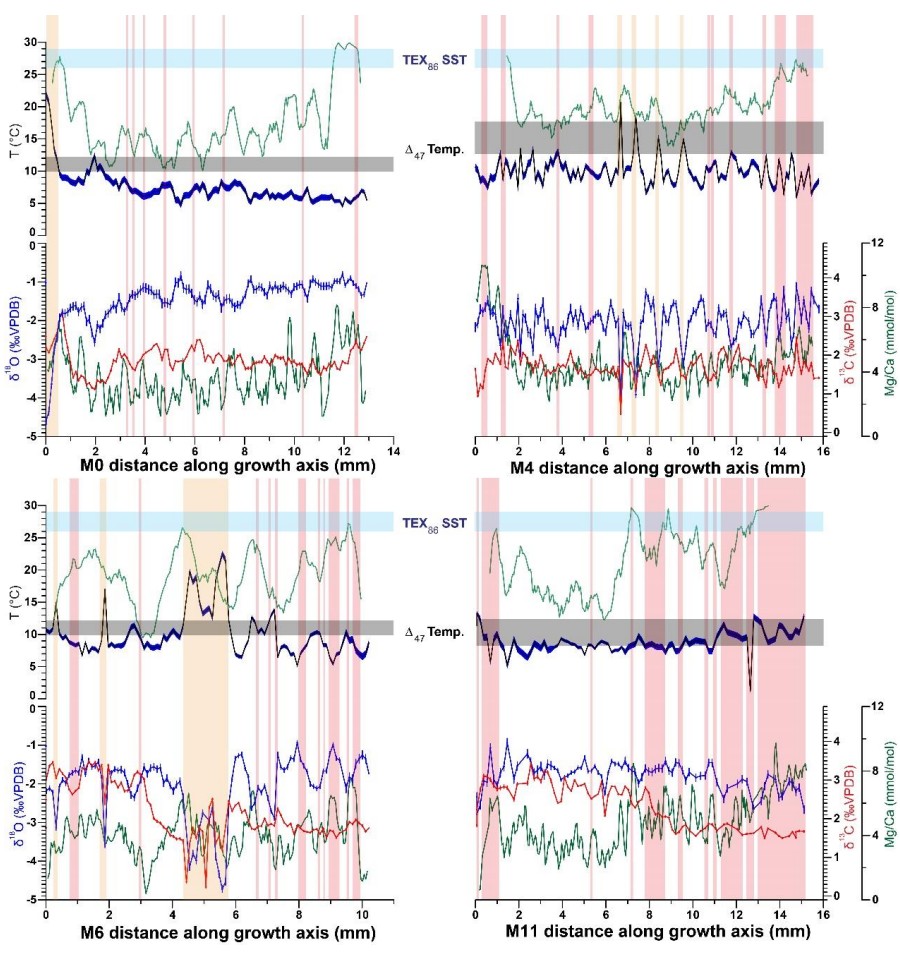

**Figure 9**

Overview of stable isotope and Mg/Ca records (bottom) as well as tentative temperature and salinity reconstructions (top) based on $\delta^{18}O$ (blue) and Mg/Ca (green), clumped isotope analysis (grey bars) and $TEX_{86}^H$ palaeothermometry (light blue bars). Temperatures calculated from $\delta^{18}O$ records (dark blue on top) are based on the calibration by Kim and O'Neil (1997) and the $\delta^{18}O_{sw}$ value of the clumped isotope measurements indicated in grey. Mg/Ca temperatures (green line on top) were calculated using the calibration reported in Surge and Lohmann (2008) with a factor 3.3 correction for lower Mg/Ca ratios in late Cretaceous ocean water. Temperatures of bulk samples of shells M4 and M11 measured using clumped isotope analysis are indicated by grey bars in graphs of M0 and M6 represent average clumped isotope temperatures of all pristine shell samples (see **Table 1**). Red and orange vertical bars indicate intervals were vesicular calcite was incorporated in the stable isotopic measurements (see **Figure 5**).





| Shell name | Sampling Location | N | δ13Cav (VPDB) ±1σ | | δ13C_record (VPDB) ±season | | δ18Oav (VPDB) ±1σ | | δ18O_record (VPDB) ±season | | D47av ±1σ | | T_av (°C) ±1σ | | δ18Osw ±1σ | |
|---|---|---|---|---|---|---|---|---|---|---|---|---|---|---|---|---|
| M0 | Shell hinge | | | | 1.91 | ±0.38 | | | -1.43 | ±0.35 | | | | | | |
| M4 | Shell hinge | 3 | 1.74 | ±0.10 | 1.73 | ±0.32 | -2.42 | ±0.12 | -1.99 | ±0.72 | 0.725 | ±0.008 | 15.2 | ±2.6 | -2.1 | ±0.7 |
| M5 | Shell hinge | 3 | 1.70 | ±0.06 | | | -2.34 | ±0.13 | | | 0.746 | ±0.016 | 9.0 | ±4.9 | -3.4 | ±1.2 |
| M6 | Shell hinge | | | | 2.28 | ±0.23 | | | -1.88 | -±0.31 | | | | | | |
| M8 | Shell hinge | 4 | 1.66 | ±0.02 | | | -1.75 | ±0.06 | | | 0.741 | ±0.008 | 10.3 | ±2.5 | -2.5 | ±0.6 |
| M11 | Shell hinge | 4 | 2.25 | ±0.08 | 2.40 | ±0.34 | -2.58 | ±0.11 | -1.74 | ±0.30 | 0.741 | ±0.007 | 10.3 | ±2.1 | -3.3 | ±0.6 |
| M5 | Ventral margin | 4 | 0.93 | ±0.15 | | | -4.36 | ±0.23 | | | 0.699 | ±0.007 | 23.8 | ±2.5 | -2.2 | ±0.7 |
| M8 | Ventral margin | 4 | -0.53 | ±0.10 | | | -7.45 | ±0.32 | | | 0.707 | ±0.012 | 21.3 | ±4.0 | -5.9 | ±1.1 |
| M10 | Ventral margin | 3 | 2.07 | ±0.34 | | | -2.99 | ±0.23 | | | 0.696 | ±0.022 | 25.4 | ±7.7 | -0.6 | ±1.8 |
| | | | | | | | | | | | | | | | | |
| Average | Shell hinge | 14 | | | | | | | | | 0.738 | ±0.004 | 11.1 | ±1.2 | -2.8 | ±0.6 |
| Average | Ventral margin | 11 | | | | | | | | | 0.643 | ±0.007 | 23.3 | ±2.9 | -3.1 | ±2.5 |

**Table 1**

Overview table of stable and clumped isotope results in this study. Rows highlighted in red represent samples from the ventral margin of the shells (which contain vesicular calcite). Rows with a white background represent samples of the dense foliated shell hinge. Note that for some shells (M5 and M8) both the ventral margin and the shell hinge was measured. Columns labelled "$\delta^{13}C$_record" and "$\delta^{18}O$_record" contain averages of the high-resolution stable isotope records measured in the shell hinges (if available, Figure 5). The bottom two rows contain average $\Delta_{47}$ and $\delta^{18}O_{sw}$ values of shell hinge (white) and ventral margin (red) samples, highlighting the difference between the two sampling strategies.