# Peer review of "An assessment of latest Cretaceous *Pycnodonte vesicularis* (Lamarck, 1806) shells as records for"

_Climate of the Past, 2017_

## Referee Comment (RC1) · Anonymous Referee #1 · 7 Nov 2017

In the manuscript "An assessment of latest Cretaceous Pycnodonte vesicularis (Lamarck, 1806) shells as records for palaeoseasonality: A multi-proxy investigation", de Winter and co-authors report observations of shell preservation and geochemistry of Pycnodonte vesicularis and potential implications for palaeoclimate and palaeoenvironmental research that can be drawn from these results.

The authors advocate that, based on conventional oxygen isotope data, Mg/Ca ratios and clumped oxygen isotopes, P. vesicularis of the late Maastrichtian of the Neuquén Basin experienced limited annual seasonality with temperatures fluctuating around

11°C or slightly more. These temperature estimates are markedly lower than existing TEX86 estimates. Additionally, general suitability of P. vesicularis for palaeoclimate research, water mass stratification and fresh water input into the Neuquén Basin are discussed.

The authors present a rigorously constrained, extensive dataset of high quality and remain generally cautious about interpretation of the data. The text, figures and tables are clear and easy to follow, even though the text is relatively long. The questions addressed are in the scope of CP and this study contains a wealth of novel data using partly very recently developed analytical techniques. Scientific methods are clearly outlined and valid, even though I partly disagree with interpretations in detail (see below). Description of the methodology is mostly sufficient to understand the workflows (see specific comments below). It is great to see that most raw data generated to write this manuscript is included in the supplements, but giving the reader some guidance to the significance of the data and more intuitive headers in the excel file would be useful. Could stable isotope ratio and clumped isotope data also be included in the supplements? In my opinion, after moderate revision, this contribution would be very suitable for Climate of the Past:

The one point I am struggling with is the inference that the oyster-based temperature estimate of 11°C can be reconciled with the 27°C TEX86 SST estimate. The authors acknowledge that the discrepancy is surprisingly large, putting forward that 1) TEX86 may be biased towards summer SST, 2) oysters are benthic creatures and bottom water temperatures at 50-75m depth would have been somewhat lower, 3) oyster growth may have been biased towards preferential shell formation in the cold season, 4) there may be an unconstrained bias on the oxygen isotope temperature estimate and Mg-based temperature estimates might be more accurate. I do fully agree with 1) and 2) even though the inferred SST-bottom water temperature gradient would be very large. In particular – if TEX86 may be biased towards high summer temperatures and oyster calcite towards winter lows, why is the seasonality recorded in the oysters so limited?

After all the authors put forward an interpretation of continuous oyster growth over the entire year with somewhat reduced $\delta$18O values only in the austral spring (October-December; Fig. 9) 3) It appears odd to me that oysters should have preferentially grown in the cold season. Modern oysters shut down growth in the cold season and show increased growth and fitness in warm temperatures (e.g., Pauley, 1988 for a review of the older literature). Average oxygen isotope values for oyster transects of modern specimens therefore show a bias towards warm temperatures (more than 5°C in a specimen from N Germany, Ullmann et al., 2010) and characteristic saw-tooth patterns with flat summer minima and sharp winter maxima of $\delta$18O (Ullmann et al., 2010, 2013). 4) It is my understanding that clumped isotope measurements are thought to represent palaeotemperatures and ambient water isotopic composition unaffected by vital effects or any other potential bias. It is not clear to me how these temperatures (if the clumped signal is indeed preserved perfectly) could be underestimated. It is unfortunate that no clumped isotope measurements for the M0 specimen are available as the authors argue this fossil shell is overall best preserved and should yield the most trustworthy data. Connected to the clumped isotopes, is there an estimate of maximum burial depth of the late Maastrichtian strata (maximum burial temperatures) in the Neuquén Basin? Can re-equilibration at the atomic scale be excluded with confidence? As the authors rightly point out there are problems with identifying a suitable transfer function for Mg/Ca temperature reconstruction for Cretaceous oysters because of secular change of seawater Mg/Ca and a multitude of available oyster (and related species) calibrations. Any argument relating to such a tentative reconstruction based on the calibration that "appears to fit best" must therefore carry some element of circularity. Could the authors revisit their chain of arguments and address these points?

Abstract and Conclusion seem quite long-winded and could be shortened with non-essential information being transferred into other sections or deleted. The referencing and reference list require a thorough check for consistency and missing information.

Specific points: Line 20: "the late Maastrichtian of the Neuquén Basin". At the moment

it reads as if only Maastrichtian sediments are present in Neuquén Basin. Line 43: "allowed for a tentative" Line 57: References in wrong sequence. Line 68: Another point here is the rapid secretion of such shells allowing for the high time-resolution required. Line 78: What is meant here by "long timescale reconstruction"? Line 79: Could this sentence be rephrased? I am not sure "caveat" can be used in the way it is put here. Line 86: References in wrong sequence Line 106: "Fischer von Waldheim, 1835" Line 106: "shell" instead of "shelf"? Line 113: Oysters in general grow very rapidly as compared to other calcite secreting marine animals and the Maastrichtian Pycnodonte does not seem to be an exception. Line 116: "tridacnid bivalves"? Line 119: "Al-Aasm" Line 131: Here and in the following, please be consistent in the use of "paleo" or "palaeo" Line 164: Missing space after 2° Line 196: "5m below the Cretaceous-Paleogene". Regardless of style, the spelling of "Paleogene" is fixed by the International Commission on Stratigraphy (e.g., Cohen et al., 2013). Line 226: "half shell"? Line 255: "Elderfield and Ganssen, 2000". This reference is missing in the reference list Line 263: This statement is somewhat vague. Is this meant to be with reference to the composition of the ambient seawater or the mantle fluid? Line 271: It should be kept in mind that the Sr distribution coefficient is negatively correlated with temperature (Rimstidt et al., 1998). Studies inferring a temperature control on Sr in bivalve calcite are rare and conversely point towards higher Sr/Ca in shell secreted at higher temperature (Wanamaker et al., 2008). The article cited in line 271 does not promote a Sr calibration but one for Mg. Line 307: What is the 1sd uncertainty of the Marbella marble related to? Does this mean that its composition is only known within 0.2 permil for carbon and 0.4 permil for oxygen or that this is its heterogeneity? In the former case this would impose quite a large potential bias on analyses corrected against this standard. In the latter case I wonder how the analytical reproducibility can be so much better (Line 308) than the above stated uncertainty ranges. Line 331: What is the meaning of "error" for the $\Delta 47$ measurements? Is that to be read as potential bias against other labs or is this purely a measurement uncertainty? Line 333: Is there a reference to these Santrock/Gonfiantini or Brand (Line 336) parameters that could be

cited here? Line 371: What is the evidence for diagenesis of the calcite comprising the vesicular material at this stage? Line 380: "consists" Line 391: "correspond to" Line 401: I am not entirely sure how interferences could cause noise in the XRF spectrum. An interference should cause a bias in the measurement which cannot be corrected for by applying a running mean smoothing routine. Noise should be bias-free and related only to the problems in quantifying low-amplitude signals precisely. Line 403: Please check for grammar. Line 454: The finding of seawater $\delta$18O values around -2 ‰be quite important. Previously some late Maastrichtian freshening of the Basin has been mentioned. Is a rough estimate of salinity possible from the reconstructed bottom water oxygen isotope ratio? Line 462: Recrystallization is a different process than cementation. This statement seems to be in contrast to what has been said in Line 371. Line 477: "laminae" Line 505: Consider adding that this is a threshold for both Mn and Fe for clarity. Line 522: "LMC" is never used again in the text so I do not think there is a need to introduce this abbreviation. Line 536: "exceedingly" seems a slightly extreme term to use. Compared to heavily altered calcite samples the ones reported here are moderately depleted in 18O. Line 540: This concept of "remote biomineralisation" has been commented on by a few studies but I am not sure how much acceptance it currently has. Line 558: The partially to fully (?) altered samples subjected to clumped isotope measurements may yield some interesting information about the type of diagenesis the samples underwent. Is there any meaningful information about burial conditions during recrystallization that can be extracted from these data? Line 595: Here and Line 596 – "Quaternary" Line 611: "altered vesicular calcite in the shell" Line 651: References in wrong sequence. Line 664: "alternating" Line 667: "correlate with" Line 677: "extra-pallial fluid" Line 693: Is this meant to be a reference to Wisshak et al. (2009)? See also lines 704, 705. Line 704: Wisshak et al. (2009) report a minor (0.5 ‰ enrichment of 18O in N. zibrowii on the basis of the Anderson and Arthur (1983) oxygen isotope thermometer. I would not count this as a strong vital effect because their assessment would have been the opposite (enrichment of 16O of similar magnitude) if they had employed the Coplen (2007) oxygen isotope thermometer. At the sub-permil level it

is very hard to make strong inferences about kinetic isotope effects. Line 705: References in wrong sequence. Line 731: "evaporitic setting"? Or "setting characterized by common evaporites"? Line 780: If Sr/Ca was indeed controlled by growth rate and P. vesicularis would have grown more slowly in spring (why would this be the case?), this effect should be seen as an ontogenetic drift of Sr/Ca towards lower ratios as the shell extends more slowly as the animal ages. The only shell that may show this effect is potentially M11, however (Fig. 5). Line 823: The way it is expressed here is potentially misleading. The Mouchi et al. (2013) calibration is not only a calibration based on juvenile specimens, it is also reported as a calibration that can only be employed for juvenile specimens. Line 847: I am not entirely sure how the estimate of 20 psu was derived. Could this be elaborated on? If this model is based on water $\delta18O$ it must depend on the isotopic composition of the fresh water source which I suppose is poorly constrained? Line 901: Secondary carbonates may be enriched in Mn and Fe and depleted in 13C and 18O, but this is not necessarily always the case. Line 916: This seems to be a repetition of lines 886 and following. Line 937: "Maastrichtian of the Neuquén Basin" Line 956: I agree that the clumped isotope numbers appear to make some sense, but is there any independent evidence that they truly reflect environmental conditions at the time of shell formation? I have the feeling that clumped isotope values are accepted once they give values reasonably close to where one would expect them and reconstructed ambient water composition is not too far off the expected marine value. Line 970: Please critically revise the reference list for typographic errors, wrong reference sequence, missing information, italics of biological species, dash length for page numbers, and superscripts of isotopic masses. A list of errors encountered during a broad check is given below. Line 974: page numbers missing Line 979: Check for upper and lower case Line 992: Brand and Veizer (1981) is not cited in the text. Line 994: page numbers missing Line 1006: Only give initials for author given names Line 1012: doi or article/page numbers missing Line 1016: doi or article/page numbers missing Line 1024: volume and page numbers missing Line 1027: volume and page numbers missing Line 1030: volume and page numbers missing Line 1053: Elderfield

and Ganssen (2000) is not referenced here bust cited in the text. Line 1060: delete "PINNA NOBILIS RATIO PROFILES" Line 1061: replace "n/a" page numbers by article number. Line 1065: This discussion paper has been followed by a revise paper in Bio-geosciences, which I think should be cited here. Line 1068: author names should not be all capitals Line 1078: Gillikin articles are in wrong sequence. Line 1079: doi missing Line 1106: Is there a final article in Climate of the Past which could be cited here instead? Line 1114: Is "Rasnnussen" indeed correct? It appears to be an error that could have been introduced by scanning and interpreting text from a paper hardcopy. Line 1155: Check reference style. Line 1159: Lorrain papers in wrong sequence. Line 1166: Information on the publication incomplete. Line 1185: "vesicularis" Line 1186: What kind of publication is "Geologie L4"? I cannot find it online. Line 1193: "Cos-mochimica" Line 1212: "Acta Palaeontologica Polonica"; page numbers missing. Line 1242: Steuber papers in wrong sequence Line 1250: Surge papers in wrong sequence Line 1262: "Rickaby, R.E.M." Line 1268: The papers cited as "a" and "b" seem to be exactly the same. Line 1280: Information on the publication details of this short course are missing. Line 1284: Please check punctuation of this reference. Line 1290: page numbers missing Line 1304: volume number missing

Figure 1: Line 1313 "A) Photograph"; "D) Lithology". Could a legend explaining the lithological signatures be included in panel D? The caption of the y-axis in panel D should read "boundary" Figure 3: Why are the sum of Mn and Fe shown in panel be? Fe is a quenching element and Mn an activator of cathodoluminescence, so I would expect that an image of Mn only would more closely resemble the CL pattern. As it stands there seems very little communality between the CL and the XRF trace which is a bit surprising. Could the small panels (C-I) be enlarged? I find it very hard to see the blocky calcite crystals in C and the thin layer of vesicular calcite in G in print. Also I do not find panel I) very convincing as evidence for a Fe and Mn corona around a boring. This boring rather seems to be Mn and Fe depleted. Figure 4E): How was the porograph constructed? Does it present porosity strictly on the pixels covered by the red arrow or does it integrate pixels in the depth domain or even pixels in depth

and with? Figure 6): Please rephrase "Cross plots showing cross plots". A): What this plot shows is a weak covariation of Mn with Fe, not that there is a link to diagenesis. This is an – admittedly well-founded – inference independent of this graph. C): This graph shows that there is no significant correlation of oxygen isotope ratios with Mn concentrations ($p > 0.05$). This contradicts the caption for this panel. In particular, most $\delta18O$ values < -3 ‰ actually seem to be related to relatively low Mn (and Fe) concentrations. Figure 7: Axis title for y-axis of panel A should be "$\Delta47$". Consider cutting the repetition of the symbol explanation in the caption for B) and state "Symbols as in A)". Figure 8): Line 1389: "Stack of proxy records for shell M0". Line 1402: A lot of the samples for which the Kim and O'Neil thermometer is employed yield results outside the calibration range (10-40°C). Consider opting for a different thermometer. Table 1: I do not understand how the average $\Delta47$ value of 0.643 was calculated. The values given above should equate to $\sim$0.701.

References:

Anderson, T. F., and Arthur, M. A., 1983, Stable isotopes of oxygen and carbon and their application to sedimentologic and paleoenvironmental problems, in Arthur, M. A., Anderson, T. F., Kaplan, I.R., Veizer, J., and Land, L. S., eds., Stable isotopes in sedimentary geology: Society of Economic Paleontologists and Mineralogists Short Course 10, p. 1.1–1.151. Cohen, K.M., Finney, S.C., Gibbard, P.L., Fan, J.-X., 2013, The ICS International Chronostratigraphic Chart. Episodes 36 (3), 199-204. Coplen, T.B., 2007, Calibration of the calcite-water oxygen-isotope geothermometer at Devils Hole, Nevada, a natural laboratory. Geochimica et Cosmochimica Acta 71, 3948-3957. Mouchi, V., de Rafélis, M., Lartaud, F., Fialin, M., Verrecchia, E., 2013, Chemical labelling of oyster shells used for time-calibrated high-resolution Mg/Ca ratios: A tool for estimation of past seasonal temperature variations. Palaeogeography, Palaeoclimatology, Palaeoecology 373, 66-74. Pauley, G. B., Van Der Raay, B., Troutt, D., 1988, Species profiles: Life histories and environmental requirements of coastal fishes and invertebrates (Pacific Northwest)—Pacific oyster, U.S. FishWildl. Serv. Biol. Rep.

82(11.85), U.S. Army Corps of Engineers, TR EL-82.4., Research and Development, National Wetlands Research Center, Washington, DC, 20240, 28 pp. Ullmann, C.V., Wiechert, U., Korte, C., 2010, Oxygen isotope fluctuations in a modern North Sea oyster (Crassostrea gigas) compared with annual variations in seawater temperature: Implications for palaeoclimate studies. Chemical Geology 277, 160-166. Ullmann, C.V., Böhm, F., Rickaby, R.E.M., Wiechert, U., Korte, C., 2013, The Giant Pacific Oyster (Crassostrea gigas) as a modern analog for fossil ostreoids: Isotopic (Ca, O, C) and elemental (Mg/Ca ,Sr/Ca, Mn/Ca) proxies. Geochemistry, Geophysics, Geosystems 14 (10), doi: 10.1002/ggge.20257. Rimstidt, J.D., Balog, A., Webb, J., 1998. Distribution of trace elements between carbonate minerals and aqueous solutions. Geochimica et Cosmochimica Acta 62 (11), 1851-1863. Wanamaker Jr, A.D., Kreutz, K.J., Wilson, T., Borns Jr, H.W., Introne, D.S., Feindel, S., 2008. Experimentally determined Mg/Ca and Sr/Ca ratios in juvenile bivalve calcite for Mytilus edulis: implications of paleotemperature reconstructions. Geo-Mar Lett 28, 359-368. Wisshak, M., López Correa, M., Gofas, S., Salas, C., Taviani, M., Jakobsen, J., Freiwald, A., 2009, Shell architecture, element composition, and stable isotope signature of the giant deep-sea oyster Neopycnodonte zibrowii sp. N. from the NE Atlantic. Deep-Sea Research I 56, 374-407.

---

## Referee Comment (RC2) · Anonymous Referee #2 · 17 Jan 2018

The manuscript "An assessment of latest Cretaceous Pycnodonte vesicularis (Lamarck, 1806) shells as records for palaeoseasonality: A multi-proxy investigation" of de Winter and coauthors wants to assess the potential of shells of the bivalve Pycnodonte vesicularis as recorder of palaeoseasonality. They analyzed several specimens coming from the late Maastrichtian Neuquén Basin in Argentina, using different techniques to check the preservation of the shells (CT scanning, light microscopy, Micro-XRF and cathodoluminescence) and to reconstruct the palaeoclimatic and palaeoenvironmental variations recorded by the bivalve (stable isotope, trace elements and

clumped isotope analyses). They described in great details the methodology used and deeply discussed the advantages and disadvantages of the different methods. The authors discussed in a proper way their results making comparison with recent closely related genera and with data coming from the literature, providing a huge amount of new information.

Results are reported in great detail, which causes the manuscript to be very long and often not fluid, due to the wealth of information provided. I understand the need to document and discuss in details the trend observed; however, I think that shorten the manuscript would definitely improve the reading. Part of the method descriptions can be moved to the supplementary material, as well as parts of the comparison with other species should be reduced. Also, the discussion (6.4 temperature proxies) and the conclusions should be shortened, as many times they results in a repetition of the same concepts.

The manuscript address interesting scientific questions that are within the scope of Climate of the Past, so I recommend its publication after moderate revision.

Specific comments

A) Paragraph 4.1. According to Figure 1, it seems that only one level with Pycnodonte vesicularis is found in the section. The caption specifies that only the main Pycnodonte level is shown. From this, I understand that there are more levels with Pycnodonte but this is not adequately described and clarified in the text. The authors only said that Pycnodonte specimens were collected from the upper 5 m below the Cretaceous-Paleogene boundary. Were the seven specimens analyzed coming from different levels? Some of the differences the authors observed among the specimens may be due to the fact they did not live during the same time interval, thus not experiencing the same environmental oscillations. Also, it is worth to add something about the taphonomic condition of the specimens (e.g., articulated, disarticulated) and the associated fauna, if present.

B) Lines 436-437 and 449. A salinity decrease by fresh water input can also cause the low $\delta18O$ and $\delta13C$ values observed, lowering both the $\delta18O$ and $\delta13C$ values (fresh water is enriched in 16O and 12C) (Gillikin, 2005; Gillikin et al., 2006). The authors should add a sentence on this and better explain why they excluded the salinity effect. Lines 436-437, add a reference to: "Such a relationship between $\delta18O$ and $\delta13C$ has often been interpreted as a sign of diagenetic alteration."

C) Paragraphs 6.3.1-6.3.3 (mainly lines 703-704). When comparing the isotope values of P. vesicularis with related species, the authors have to take in mind the different environmental settings in which the 3 species live (P. vesicularis, N. zibrowii and H. hyotis). As observed by the authors N. zibrowii lives in deep water, so its isotope signatures (especially the $\delta18O$ values) are also controlled by this parameter. The higher $\delta18O$ values recorded in N. zibrowii compared to P. vesicularis may be also explained with the deep sea habitat of the former species. So if they want to compare the isotope values, they have to consider species coming from similar environments.

D) Lines 709-712. This sentence is strange; are you sure is the juvenile and not the adult part of the shells showing an ontogenetic trend in $\delta13C$? Usually bivalves incorporates isotopically light CO2 in the adult stages, showing an ontogenetic decrease in $\delta13C$ (e.g., Gillikin, 2005; Gillikin et al., 2006, 2007). The model of Lorrain et al. (2004) suggests that the decrease in $\delta13C$ through ontogeny is actually caused by increasing utilization of metabolic C (respiratory CO2 which is 13C-depleted) to satisfy carbon requirements for calcification. As bivalves grow and become older, the amount of available metabolic CO2 increases, while the amount needed for shell growth is reduced, resulting in more metabolic carbon (12C-enriched) being incorporated into the shell. A similar ontogenetic trend is observed in specimens M6 and M11. The authors should rewrite this part.

E) The authors provide a lot of data in the manuscript, analyzing in details the different methods used. I understand that the primary aim of the manuscript is to assess the potential of P. vesicularis shells as recorders of palaeoseasonality. However, the

authors obtained some useful data for palaeoclimatic reconstructions which are not adequately discussed in the manuscript. How the data in terms of palaeotemperatures and palaeoseasonality fit into the larger context of the Cretaceous climate of the area? Which new information can they add to the knowledge of the late Cretaceous of the Neuquén basin?

Minor comments

A) Be consistent through the manuscript on the use of English or American spelling (paleo -> palaeo, recrystallization, recrystallised, . . .)

B) When citing a paper within the manuscript use the same format. Some citations have comma before the year other not, e.g., Kiessling et al. 2005 (line 177) or Woelders et al., 2017 (line 182). Check carefully through the text.

LINE 68-70 Bivalve shells are also important as they have a broad biogeographic distribution, occurring in different environmental settings, from shallow water to deep-sea environments, in freshwater, marine and brackish settings, from near the poles to the equator (e.g., Schöne et al., 2005a) LINE 78 Add other references as Schöne et al., 2005b; Butler et al., 2013 LINES 92-96 Add reference to Crippa et al., 2016 LINE 111 ReconstructionS LINE 130 "The aim of this multi-approach is to characterize the MICROstructure". Refer also in other part of the text to microstructure and not structure, as you are observing shells at micrometrical scale LINE 196 "from the upper 5 m OF below the Cretaceous". Delete OF LINE 200 What do you mean by biodegradation? Please explain LINE 242 It is not Figure 1, please correct LINE 244 "See section 4.1.1 and 4.1.3". May it be section 5? LINES 252 and 274 Gillikin et al., before Lorrain et al. LINE 257 Surge and Lohmann 2008 before Wanamaker et al. 2008 LINE 294 Add space between 100 and $\mu$m LINE 345 Add reference to MacDonald et al., 2009 LINE 371 Diagenetic alteration instead of diagenesis LINES 384-385 and 476-477 What about the CL of the vesicular layer? Add an image of this; if not in the main paper, add more images in the supplementary. It is important to document what you

saw and described. LINE 401 Delete space after record LINES 401-403 Rephrase this sentence LINE 405 "In three out of four specimens", delete OUT LINE 445 What does it mean from the same locality? Same stratigraphic level? LINE 446 Defliese et al., 2015; the year should be in parentheses LINES 471-474 Also, oystreids, due to their layered shell structure, may be more prone to infiltration of fluids inside the shells, which of course affected more the porous chalky fabric than the foliated ones. LINE 477 Laminae instead of lamina LINE 521 Measuring instead of measured LINE 566 "in vesicular calcite this close". Delete THIS LINE 596 ReconstructionS LINE 611 "vesicular calcite into the in the shell". Delete INTO THE LINE 651 Klein et al. before Ullmann et al. LINE 654 Delete "in the Late Cretaceous" at the end of the sentence; it is clear you are referring to the Late Cretaceous LINE 662 Hyotissa not Hytissa; add the name of the author who first describe the species LINE 683 "which complicates inter-pretation"; it should be "which complicates THE interpretation" or "which complicates interpretationS" LINE 693, 704 and other lines Wisshak et al., 2008, in the reference list is Wisshak et al., 2009 LINE 705 Titschack et al., 2010 before Ullmann et al., 2010 LINE 731 Evaporitic setting LINE 732 Specify in which country Safaga Bay is LINE 732 Add + before 2.17 ‰ LINE 734 "records OF H. hyotis". Add OF LINE 745 "a decrease in salinity in the spring". Delete THE LINE 781-782 It seems strange that during high productivity spring they growth slower, they should do the opposite. Is there any ev-idence in previous literature on this? LINE 783 Gillikin et al., 2005 before Lorrain et al., 2005 LINE 796 Such a decrease of nearly 10°C between surface and relatively deep sea water is comparable to present day situation? LINE 801 "d18O", change with $\delta$18O LINE 806-808 During the spring-summer seasons the authors reported a salinity decrease; slow growth may be caused by this? LINE 815 "than parts of the year". Add OTHER parts of the year LINE 847 How was 20 PSU determined? It is a very big variations. For example in the Mediterranean Sea a salinity change of 2 PSU would correspond to a shift of ∼1‰ in $\delta$18Osw (Rohling and Bigg, 1998), which is equivalent to nearly 4-5°C in the temperatures calculated from the $\delta$18O of the shell. The authors observed a 10°C variation, which correspond to ∼2-3‰ in $\delta$18O. The salinity 20 PSU

value seems overestimated. The authors should better explain this assertion LINE 869 10-15 °C at which water depth? LINE 909 reconstructionS LINE 911 vesicular instead of vesulicar

Reference list

Please check very carefully the reference list. Some data are missing (pages), many specific names are not in italic, some references are in wrong chronological order, some present in the list are missing in the main text and viceversa. Some of the changes to make are listed hereafter: LINE 979 Add capital letters for places and time LINE 990 Hyotissa hyotis in italic LINES 992 and 994 Switch references, wrong chronological order LINE 992 Brand and Veizer 1981 not in the main text LINE 996 Species name in italic, and vesicularis in lowercase LINES 1006 and 1009 Carriker Melbourne, one is full name the other is Carriker M. LINE 1015 Cleroux et al. not in the main text LINES 1035-1039 All Dettman's references are in the wrong chronological order LINE 1041 Wrong citation of this reference in the text; it should be Dlugokencky and Tans LINE 1078 Gillikin et al, 2005b should be moved before LINE 1140 Is de Lartaud 2000a in the main text (line 389) the same reference? Lartaud 2010a is missing in the main text LINES 1159-1161 Switch references, wrong chronological order LINE 1167 Wrong citation of this reference in the text; it should be Malumian and Nanez (line 168) LINE 1228 and 1230 Switch references, Schöne before Schouten LINES 1230 and 1232 Schöne's references missing in the text LINE 1238 Stenzel, 1956 or 1959? In the text is 1959 LINE 1240 and 1242 Switch references, wrong chronological order LINE 1244 In line 78 and other part of the text wrong reference of Steuber 2005; it should be Steuber et al., 2005 LINE 1250 Surge and Owens, 2003 should be moved before, but it is missing in the text LINE 1262 and 1265 Switch references, wrong chronological order LINES 1268-1270 Same reference repeated LINE 1295 Reference missing in the text. Also, Marali and Schone, 2014; Scourse et al., 2006 are missing in the reference list

Figures

FIGURE 1 Is it possible to add a legend with the lithologies? Also, in the y-axis of the log correct BOUNDARY FIGURE 2 To be more clear the direction of growth of the shell should be added. FIGURE 3 Images C-G and H, I should be a bit larger. Images C and E are not very clear. FIGURES 5 and 9 Vertical bars have too similar colors (orange and red), change one to be more clear. FIGURE 6 Cross plots showing cross plots, please rephrase. FIGURE 8 "interpretation of annual cyclicity based on Sr/Ca ratios" and on $\delta$18O seasonality?

References

Butler, P.G., Wanamaker Jr., A.D., Scourse, J.D., Richardson, C.A. & Reynolds, D.J. (2011), Longterm stability of 13C with respect to biological age in the aragonite shell of mature specimens of the bivalve mollusk Arctica islandica, Palaeogeography, Palaeoclimatology, Palaeoecology, 302(1), 21-30.

Crippa G., Angiolini L., Bottini C., Erba E., Felletti F., Frigerio C., Hennissen J.A.I., Leng M.J., Petrizzo M.R., Raffi I., Raineri G., Stephenson M.H. (2016). Seasonality fluctuations recorded in fossil bivalves during the early Pleistocene: Implications for climate change. Palaeogeogr. Palaeoclimatol. Palaeoecol., 446, 234–251.

Gillikin, D.P. (2005), Geochemistry of Marine Bivalve Shells: the potential for paleoenvironmental reconstruction. Unpublished PhD thesis, Bruxelles University.

Gillikin, D.P., Lorrain, A., Bouillon, S., Willenz, P. & Dehairs, F. (2006), Stable carbon isotope composition of Mytilus edulis shells: Relation to metabolism, salinity, $\delta$13CDIC, and phytoplankton, J. Organic Geochemistry, 37, 1371–1382.

Gillikin, D.P., Lorrain, A., Meng, L. & Dehairs, F. (2007), A large metabolic carbon contribution to the $\delta$13C record in marine aragonitic bivalve shells, Geochimica et Cosmochimica Acta, 71, 2936–2946.

Lorrain, A., Paulet, Y.M., Chauvaud, L., Dunbar, R., Mucciarone, D. & Fontugne, M. (2004), $\delta$13C variation in scallop shells: increasing metabolic carbon contribution with

body size?, Geochimica et Cosmochimica Acta, 68, 3509–3519.

MacDonald, J., Freer, A., & Cusack, M. (2009). Alignment of crystallographic c-axis throughout the four distinct microstructural layers of the oyster Crassostrea gigas. Crystal Growth & Design, 10(3), 1243-1246.

Rohling, E.J., Bigg, G.R. (1998). Paleosalinity and $\delta$18O: a critical assessment. J. Geophys. Res. Oceans 103 (C1), 1307–1318.

Schöne, B.R., Fiebig, J., Pfeiffer, M., Gle$\beta$, R., Hickson, J., Johnson, A.L.A., Dreyer, W. & Oschmann, W. (2005a), Climate records from a bivalved Methuselah (Arctica islandica, Mollusca; Iceland), Palaeogeography, Palaeoclimatology, Palaeoecology, 228(1), 130-148.

Schöne, B.R., Pfeiffer, M., Pohlmann, T. & Siegismund, F. (2005b), A seasonally resolved bottom water temperature record for the period of AD 1866–2002 based on shells of Arctica islandica (Mollusca, North Sea), Int. J. Climatol., 25, 947–962.

---

## Author Comment (AC1) · 14 Feb 2018

Dear Climate of the Past Editorial Board, dear Referees,

We would like to express our gratitude to both anonymous Referees for their thoughtful review of our manuscript titled "An assessment of latest Cretaceous *Pycnodonte vesicularis* (Lamarck, 1806) shells as records for palaeoseasonality: A multi-proxy investigation" submitted to the journal Climate of the Past. After careful consideration of all the criticisms and suggestions raised by the Referees, we herewith provide a point-by-point reply to the report of both Referees. In this rebuttal, we will first summarize the major points of criticism raised by the Referees and then proceed to cite parts of the review reports and provide our reactions directly below the citations.

**Major points of criticism**

1. **Offset between independent temperature reconstructions**

One of the major concerns raised by Referee #1 is the way in which our manuscript describes the offset between the various independent temperature reconstructions we attempted. The point we were trying to make in our discussion of this offset is that TEX86 temperatures seem to overestimate the water temperatures, or at least those in the part of the water column where the *Pycnodonte* shells grew. We make the case that clumped isotope thermometry should be the most sensitive and well-constrained of the methods we applied. Given that it yields lower temperatures, we conclude that it is likely that the TEX86 reconstructions are overestimating the local water temperatures. Referee #1 suggests that our clumped isotope measurements might be subject to diagenetic alteration or solid state reordering after burial, but this effect would increase the reconstructed temperature, not lower it. We include diagenetically altered clumped-isotopic values in our manuscript that illustrate this effect and indeed yield higher temperatures (26-35 degrees). In a revised version of the manuscript, we will include these altered clumped isotope measurements into our discussion as an illustration of the burial temperatures (as suggested by Referee #1). Following the Referee's suggestion, we will also clarify that we do not think that the temperature gradient with (50-75m) depth can fully explain the offset between TEX86 and clumped isotope reconstructions. Similarly, we agree that extreme summer temperatures might not fully explain this offset while still maintaining relatively cool mean temperatures and moderate seasonality. However, we do not wish to fully abandon the hypothesis that shell growth happened preferentially in the cooler months. If summer maxima in temperature or minima in salinity were extreme enough to produce conditions that limited shell growth, we reckon that this may still be a valid explanation. Indeed, our records suggest that lowest salinities are reached in winter to spring. In the Cretaceous lower mid-latitudes of the South Atlantic the growth of ostreid bivalves may not be limited by low winter temperatures, as they are in the cool temperate seas of the present North Sea Basin (NW Europe) (as in Ullmann et al., 2010,2013). It may, in our opinion, therefore not be ruled out that preferential growth in cool seasons biased reconstructions of mean annual temperatures from these bivalves towards lower temperatures. The part of the discussion that deals with this offset between temperature reconstructions will be revisited in the revised version and we will attempt to restructure the arguments to make the discussion of the points mentioned above more convincing.

2. **Stratigraphic constraint**

A point is raised over the lack of stratigraphic constraints of the samples used in our manuscript. We admit that Figure 1 could be clearer and also realize that it may be interesting to constrain the stratigraphy of the specimens more rigorously. Besides, we agree with the Referees that it may be interesting for future research to briefly state how abundant *Pycnodonte* shells are in the

stratigraphy. We therefore propose to restructure paragraph 4.1 and reconsider Figure 1 to clarify these points. In the meantime, we will include more detailed information about the taphonomic condition of the specimens. We agree that any variation or uncertainty in the stratigraphic location of the specimens may have bearing on the discussion of differences in seasonality recorded in the multi-proxy records through the shells, and will take this into account in the discussion while revising our manuscript.

**3. Vital effects and comparison with other species**

Some comments were raised with regard to the species with which we compare our *Pycnodonte* records in the manuscript. For these comparisons, we have tried to stick to the closest modern relatives of *Pycnodonte vesicularis*. We do however, concede that these modern analogues may not be a perfect fit in terms of their environment. Most notably the species *Neopycnodonte zibrowii* may not be a good modern analogue. We will reduce the discussion of the comparison with this species and focus on the discussion with *Hyotissa*, which has a more similar environment.

**Length of the manuscript text**

Both Referees rightly comment on the fact that our manuscript is quite long. We realize that the text has become quite convoluted in some places (most notably in the discussion) as a result of the various methods applied in this multi-proxy study. We thank both Referees for their advice on how to amend this and will try to use their suggestions to improve the readability of the manuscript text. We will try to shorten methodological descriptions and move parts of the methodology to supplementary files. We will shorten the Abstract and Conclusion chapters and will attempt to remove any recurring parts from the discussion to make the argumentation easier to follow.

**Point-by-point replies to comments of referees**

*Anonymous Referee #1*

*In the manuscript "An assessment of latest Cretaceous Pycnodonte vesicularis (Lamarck, 1806) shells as records for palaeoseasonality: A multi-proxy investigation", de Winter and co-authors report observations of shell preservation and geochemistry of Pycnodonte vesicularis and potential implications for palaeoclimate and palaeoenvironmental research that can be drawn from these results.*

*The authors advocate that, based on conventional oxygen isotope data, Mg/Ca ratios and clumped oxygen isotopes, P. vesicularis of the late Maastrichtian of the Neuquén Basin experienced limited annual seasonality with temperatures fluctuating around 11_C or slightly more. These temperature estimates are markedly lower than existing TEX86 estimates. Additionally, general suitability of P. vesicularis for palaeoclimate research, water mass stratification and fresh water input into the Neuquén Basin are discussed.*

*The authors present a rigorously constrained, extensive dataset of high quality and remain generally cautious about interpretation of the data. The text, figures and tables are clear and easy to follow, even though the text is relatively long. The questions addressed are in the scope of CP and this study contains a wealth of novel data using partly very recently developed analytical techniques. Scientific methods are clearly outlined and valid, even though I partly disagree with interpretations in detail (see below). Description of the methodology is mostly sufficient to understand the workflows (see specific comments below). It is great to see that most raw data*

*generated to write this manuscript is included in the supplements, but giving the reader some guidance to the significance of the data and more intuitive headers in the excel file would be useful. Could stable isotope ratio and clumped isotope data also be included in the supplements? In my opinion, after moderate revision, this contribution would be very suitable for Climate of the Past:*

*The one point I am struggling with is the inference that the oyster-based temperature estimate of 11_C can be reconciled with the 27_C TEX86 SST estimate. The authors acknowledge that the discrepancy is surprisingly large, putting forward that 1) TEX86 may be biased towards summer SST, 2) oysters are benthic creatures and bottom water temperatures at 50-75m depth would have been somewhat lower, 3) oyster growth may have been biased towards preferential shell formation in the cold season, 4) there may be an unconstrained bias on the oxygen isotope temperature estimate and Mg-based temperature estimates might be more accurate. I do fully agree with 1) and 2) even though the inferred SST-bottom water temperature gradient would be very large. In particular – if TEX86 may be biased towards high summer temperatures and oyster calcite towards winter lows, why is the seasonality recorded in the oysters so limited? After all the authors put forward an interpretation of continuous oyster growth over the entire year with somewhat reduced _18O values only in the austral spring (October- December; Fig. 9) 3) It appears odd to me that oysters should have preferentially grown in the cold season. Modern oysters shut down growth in the cold season and show increased growth and fitness in warm temperatures (e.g., Pauley, 1988 for a review of the older literature). Average oxygen isotope values for oyster transects of modern specimens therefore show a bias towards warm temperatures (more than 5_C in a specimen from N Germany, Ullmann et al., 2010) and characteristic saw-tooth patterns with flat summer minima and sharp winter maxima of _18O (Ullmann et al., 2010, 2013). 4) It is my understanding that clumped isotope measurements are thought to represent palaeotemperatures and ambient water isotopic composition unaffected by vital effects or any other potential bias. It is not clear to me how these temperatures (if the clumped signal is indeed preserved perfectly) could be underestimated. It is unfortunate that no clumped isotope measurements for the M0 specimen are available as the authors argue this fossil shell is overall best preserved and should yield the most trustworthy data. Connected to the clumped isotopes, is there an estimate of maximum burial depth of the late Maastrichtian strata (maximum burial temperatures) in the Neuquén Basin? Can re-equilibration at the atomic scale be excluded with confidence? As the authors rightly point out there are problems with identifying a suitable transfer function for Mg/Ca temperature reconstruction for Cretaceous oysters because of secular change of seawater Mg/Ca and a multitude of available oyster (and related species) calibrations. Any argument relating to such a tentative reconstruction based on the calibration that "appears to fit best" must therefore carry some element of circularity. Could the authors revisit their chain of arguments and address these points?*

As mentioned above, this is one of the major points in the discussion that we will try to clarify in a revised version of the manuscript. We will put in effort to more clearly reconcile the outcomes of various techniques of temperature reconstruction described in the manuscript.

*Abstract and Conclusion seem quite long-winded and could be shortened with nonessential information being transferred into other sections or deleted. The referencing and reference list require a thorough check for consistency and missing information.*

Both Referees have noted that our reference list is not up to date. In a revised version of the manuscript, we will take care to update the reference list according to the guidelines of Climate of

the Past. As mentioned above, we will restructure Abstract and Conclusions to make them more concise.

*Specific points:*

*Line 20: "the late Maastrichtian of the Neuquén Basin". At the moment it reads as if only Maastrichtian sediments are present in Neuquén Basin.*

We will rephrase this to "the late Maastrichtian strata exposed within the Neuquén Basin succession"

*Line 43: "allowed for a tentative"*

This will be rephrased

*Line 57: References in wrong sequence.*

We will update the reference list and citations through the text.

*Line 68: Another point here is the rapid secretion of such shells allowing for the high time-resolution required.*

We thank Referee #1 for this suggestion and incorporate this into our manuscript text.

*Line 78: What is meant here by "long timescale reconstruction"?*

We mean to refer to the superposition of seasonality records from bivalves into the framework of longer timescale palaeoclimate records. This will be rephrased to: "seasonally-resolved bivalve records are rarely combined with longer timescale palaeoclimate records"

*Line 79: Could this sentence be rephrased? I am not sure "caveat" can be used in the way it is put here.*

We will rephrase this to "disadvantage"

*Line 86: References in wrong sequence*

References and citations will be updated

*Line 106: "Fischer von Waldheim, 1835"*

This will be corrected

*Line 106: "shell" instead of "shelf"?*

The term "commissural shelf" refers to the shelflike part of the bivalve shell that faces the place where both valves meet (the commissure).

*Line 113: Oysters in general grow very rapidly as compared to other calcite secreting marine animals and the Maastrichtian Pycnodonte does not seem to be an exception.*

We will take this into account while rephrasing this sentence. In this context we mean to say that bivalves (as far as we know) seem to precipitate their shell calcite in isotopic equilibrium with the extrapallial fluid (certainly with respect to oxygen isotopes).

*Line 116: "tridacnid bivalves"?*

This will be corrected

*Line 119: "Al-Aasm"*

This will be corrected

*Line 131: Here and in the following, please be consistent in the use of "paleo" or "palaeo"*

In the revised version of the manuscript, we will go through the text and consistently use the British spelling ("palaeo") wherever applicable.

*Line 164: Missing space after 2_*

This will be corrected

*Line 196: "5m below the Cretaceous-Paleogene". Regardless of style, the spelling of "Paleogene" is fixed by the International Commission on Stratigraphy (e.g., Cohen et al., 2013).*

Agreed, this will be corrected

*Line 226: "half shell"?*

This will be corrected

*Line 255: "Elderfield and Ganssen, 2000". This reference is missing in the reference list*

We will add it to the reference list

*Line 263: This statement is somewhat vague. Is this meant to be with reference to the composition of the ambient seawater or the mantle fluid?*

We agree that this sentence could require clarification and will adapt it in the next version. What is meant is that trace element concentrations (in this case those of Mg) are not necessarily taken into the shell in equilibrium with the ambient seawater.

*Line 271: It should be kept in mind that the Sr distribution coefficient is negatively correlated with temperature (Rimstidt et al., 1998). Studies inferring a temperature control on Sr in bivalve calcite are rare and conversely point towards higher Sr/Ca in shell secreted at higher temperature (Wanamaker et al., 2008). The article cited in line 271 does not promote a Sr calibration but one for Mg.*

We apologize for the mixing up of references in this sentence and will revisit this paragraph to provide the right background of Sr/Ca ratios in bivalve shells and how they should be interpreted.

*Line 307: What is the 1sd uncertainty of the Marbella marble related to? Does this mean that its composition is only known within 0.2 permil for carbon and 0.4 permil for oxygen or that this is its heterogeneity? In the former case this would impose quite a large potential bias on analyses corrected against this standard. In the latter case I wonder how the analytical reproducibility can be so much better (Line 308) than the above stated uncertainty ranges.*

This is a valid comment, as the description of the MAR2 standard contains errors. The uncertainty on the values of the Marbella marble is relative to the values of NBS-19 and the reported values should be of 2 standard deviations rather than one. A combination of machine error and reproducibility error on the MAR2 standard yields total uncertainties on the measurement of 0.1 and 0.2 permille for carbon and oxygen delta values respectively. We will clarify this in the revised text to avoid confusion.

*Line 331: What is the meaning of "error" for the _47 measurements? Is that to be read as potential bias against other labs or is this purely a measurement uncertainty?*

This error is a long-term measurement reproducibility error. We will mention this to avoid confusion.

*Line 333: Is there a reference to these Santrock/Gonfiantini or Brand (Line 336) parameters that could be cited here?*

Two references for these parameters are:

Daëron, M. et al., 2016. Absolute isotopic abundance ratios and the accuracy of Δ47 measurements. Chemical Geology, 442, pp.83–96.
Schauer, A.J. et al., 2016. Choice of 17O correction affects clumped isotope (Δ 47) values of CO 2measured with mass spectrometry. Rapid Communications in Mass Spectrometry, 30(24), pp.2607–2616.

We will include the reference to these studies in the revised version. In order to clarify and shorten the method description in this paragraph we will move some of the text to supplementary material.

*Line 371: What is the evidence for diagenesis of the calcite comprising the vesicular material at this stage?*

This is a valid point, in the revised version of the manuscript we will refrain from interpreting results in the results section and save interpretations for the discussion section.

*Line 380: "consists"*

This will be corrected

*Line 391: "correspond to"*

This will be corrected

*Line 401: I am not entirely sure how interferences could cause noise in the XRF spectrum. An interference should cause a bias in the measurement which cannot be corrected for by applying a running mean smoothing routine. Noise should be bias-free and related only to the problems in quantifying low-amplitude signals precisely.*

This is a valid point and we realize that the explanation of smoothing of the XRF records is somewhat vague as it is. In the revised version of the manuscript we will rephrase this sentence to explain that small-scale variations in the matrix of the sample (in this case the bivalve shell) causes variations in the spectral resolution of XRF spectra, which affect lighter elements with smaller peaks (e.g. Mg) more than elements with larger peaks in the XRF spectrum.

*Line 403: Please check for grammar.*

This sentence will be corrected

*Line 454: The finding of seawater _18O values around -2 ‰be quite important. Previously some late Maastrichtian freshening of the Basin has been mentioned. Is a rough estimate of salinity possible from the reconstructed bottom water oxygen isotope ratio?*

We will try to include an estimate of salinity changes in the Basin based on our data, but this will be discussed in the Discussion section.

*Line 462: Recrystallization is a different process than cementation. This statement seems to be in contrast to what has been said in Line 371.*

We agree and will rephrase by stating which process may affect the porosity of the shell on the microscale.

*Line 477: "laminae"*

This will be corrected

*Line 505: Consider adding that this is a threshold for both Mn and Fe for clarity.*

We will follow the suggestion and add a statement here.

*Line 522: "LMC" is never used again in the text so I do not think there is a need to introduce this abbreviation.*

Agreed, we will remove the reference to the abbreviation.

*Line 536: "exceedingly" seems a slightly extreme term to use. Compared to heavily altered calcite samples the ones reported here are moderately depleted in 18O.*

Agreed, we will use "relatively" instead.

*Line 540: This concept of "remote biomineralisation" has been commented on by a few studies but I am not sure how much acceptance it currently has.*

Neither are we, but we would like to include the reference to this hypothesis for sake of completeness.

*Line 558: The partially to fully (?) altered samples subjected to clumped isotope measurements may yield some interesting information about the type of diagenesis the samples underwent. Is there any meaningful information about burial conditions during recrystallization that can be extracted from these data?*

We thank Referee #1 for this suggestion, which, as mentioned above, will be included in a revision of the discussion. Indeed, diagenetically altered clumped isotope results may feature more prominently in this part of the discussion to show that the samples considered reliable are not affected by the same degree of diagenesis.

*Line 595: Here and Line 596 – "Quaternary"*

This will be corrected

*Line 611: "altered vesicular calcite in the shell"*

This will be corrected

*Line 651: References in wrong sequence.*

This will be corrected

*Line 664: "alternating"*

This will be corrected

*Line 667: "correlate with"*

This will be corrected

*Line 677: "extrapallial fluid"*

This will be corrected

*Line 693: Is this meant to be a reference to Wisshak et al. (2009)? See also lines 704, 705.*

This will be corrected

*Line 704: Wisshak et al. (2009) report a minor (0.5 ‰ enrichment of 18O in N. zibrowii on the basis of the Anderson and Arthur (1983) oxygen isotope thermometer. I would not count this as a strong vital effect because their assessment would have been the opposite (enrichment of 16O of similar magnitude) if they had employed the Coplen (2007) oxygen isotope thermometer. At the sub-permil level it is very hard to make strong inferences about kinetic isotope effects.*

We will take this into account while discussing differences between our results and those of Wisshak et al. 2009. As mentioned above, we will limit the discussion of *N. zibrowii* as a modern analogue because its environment is not likely to be representative for that of *P. vesicularis*.

*Line 705: References in wrong sequence.*

This will be corrected

*Line 731: "evaporitic setting"? Or "setting characterized by common evaporites"?*

This will be corrected

*Line 780: If Sr/Ca was indeed controlled by growth rate and P. vesicularis would have grown more slowly in spring (why would this be the case?), this effect should be seen as an ontogenetic drift of Sr/Ca towards lower ratios as the shell extends more slowly as the animal ages. The only shell that may show this effect is potentially M11, however (Fig. 5).*

We thank Referee #1 for the input on this part of the discussion and will take these considerations into account in the revision of our discussion. We acknowledge that the relationship of Sr/Ca with growth rate does not show up as an ontogenetic trend. Sr/Ca ratios in bivalve calcite are controversial and our current interpretation of these records is based on a small amount of earlier studies. As it stands, the interpretation of Sr/Ca ratios was mostly driven by their anti-phase relationship with carbon isotopes, which were tentatively interpreted as proxy for light conditions based on the comparison with *Hyotissa*. As mentioned above, we may have to partly revise and restructure our interpretation of the phase relationship between the proxies. Therefore, we will revise the discussion of Sr/Ca ratios in the new version of the manuscript and be more careful in interpreting Sr/Ca ratios. We will put forward other interpretations for this record and its apparent seasonality to show that the relationship with growth rate is not certain.

*Line 823: The way it is expressed here is potentially misleading. The Mouchi et al. (2013) calibration is not only a calibration based on juvenile specimens, it is also reported as a calibration that can only be employed for juvenile specimens.*

We will rephrase this expression to avoid confusion. The attempted temperature reconstructions using Mg/Ca ratios are subject to considerable uncertainty. We will shorten the discussion of these Mg/Ca temperatures and further emphasize that these reconstructions are based on assumptions that are uncertain. The reason these attempted temperature reconstructions are added is to illustrate how different independent reconstructions lead to different temperatures and to allow discussion between temperature proxies.

*Line 847: I am not entirely sure how the estimate of 20 psu was derived. Could this be elaborated on? If this model is based on water _18O it must depend on the isotopic composition of the fresh water source which I suppose is poorly constrained?*

Both referees mention this issue. We acknowledge that the estimate of salinity is not very certain. We will rephrase this sentence in the revised version and refrain from giving absolute estimates of salinity since we cannot constrain all the variables needed to do so with any certainty. Instead, we will discuss relative salinity changes.

*Line 901: Secondary carbonates may be enriched in Mn and Fe and depleted in 13C and 18O, but this is not necessarily always the case.*

We will rephrase this to "recrystallization and precipitation of secondary carbonates in equilibrium with these reducing pore fluids may raise the concentration of Mn and Fe and lower stable isotope ratios"

*Line 916: This seems to be a repetition of lines 886 and following.*

Agreed, we will shorten this paragraph and remove repetitive elements to improve readability of the text.

*Line 937: "Maastrichtian of the Neuquén Basin"*

This will be corrected

*Line 956: I agree that the clumped isotope numbers appear to make some sense, but is there any independent evidence that they truly reflect environmental conditions at the time of shell formation? I have the feeling that clumped isotope values are accepted once they give values reasonably close to where one would expect them and reconstructed ambient water composition is not too far off the expected marine value.*

In the revised manuscript we will revisit the discussion of clumped isotope results and discuss any potential for diagenetic alteration in more detail using the altered clumped isotope measurements that yield higher temperatures.

All comments and suggestions regarding the reference list were incorporated in the revised manuscript.

*Figure 3: Why are the sum of Mn and Fe shown in panel be? Fe is a quenching element and Mn an activator of cathodoluminescence, so I would expect that an image of Mn only would more closely resemble the CL pattern. As it stands there seems very little communality between the CL and the XRF trace which is a bit surprising. Could the small panels (C-I) be enlarged? I find it very hard to see the blocky calcite crystals in C and the thin layer of vesicular calcite in G in print. Also I do not find panel I) very convincing as evidence for a Fe and Mn corona around a boring. This boring rather seems to be Mn and Fe depleted.*

This figure and its caption will be restructured to better highlight the observations they are meant to illustrate. The corona of higher Mn and Fe concentrations is maybe better visible in the larger image Figure 3B.

*Figure 4E): How was the porograph constructed? Does it present porosity strictly on the pixels covered by the red arrow or does it integrate pixels in the depth domain or even pixels in depth and with?*

Details on the processing of CT-scan data will be added to the part of the methods section that will be placed in supplementary data to allow for easier reading. The porographs are constructed by first segmenting the scanned object into two phases, i.e., shell material and porosity. Porographs are made by integrating density data parallel to the shell layers:

The segmentation is straightforward, given the large density difference between the shell material and the pore filling substance (air, Araldite resin did not penetrate into the internal pores). Secondly, the porosity is calculated (in Matlab, repetitive 2D approach) by taking the slice per slice ratio of pore pixels to pore + shell pixels (pore/pore+shell). The porograph shows the evolution of porosity from the bottom to the top slice in a given image stack. The slice per slice calculation allows to evaluate the evolution of porosity through the shell. The total shell porosity is then obtained by taking the average porosity of all values that were obtained in the slice per slice procedure. The calculated total shell porosity was confirmed by a voxel based volume approach (in Avizo fire, in 3D) that takes into account the total volume of pore and shell voxels.

*Figure 6): Please rephrase "Cross plots showing cross plots". A): What this plot shows is a weak covariation of Mn with Fe, not that there is a link to diagenesis. This is an – admittedly well-founded – inference independent of this graph. C): This graph shows that there is no significant correlation of oxygen isotope ratios with Mn concentrations (p > 0.05). This contradicts the caption for this panel. In particular, most _18O values < -3 ‰ actually seem to be related to relatively low Mn (and Fe) concentrations.*

We will rephrase the figure captions and better describe the trends shown in this figure. In addition, we will base our discussion of diagenesis in the records more on Figure 5, which shows the effect of local diagenesis on the records more clearly.

*Figure 7: Axis title for y-axis of panel A should be "_47". Consider cutting the repetition of the symbol explanation in the caption for B) and state "Symbols as in A)".*

We thank the referee for this suggestion and will implement it in the revised version.

*Figure 8):*

*Line 1389: "Stack of proxy records for shell M0".*

This will be corrected

*Line 1402: A lot of the samples for which the Kim and O'Neil thermometer is employed yield results outside the calibration range (10-40_C). Consider opting for a different thermometer.*

We will attempt a different thermometer to accommodate the samples with higher oxygen isotope values (lower temperatures).

*Table 1: I do not understand how the average _47 value of 0.643 was calculated. The values given above should equate to _0.701.*

Indeed it seems that this is an error in the table which will be corrected in the revised version of the manuscript.

References:

Anderson, T. F., and Arthur, M. A., 1983, Stable isotopes of oxygen and carbon and their application to sedimentologic and paleoenvironmental problems, in Arthur, M. A., Anderson, T. F., Kaplan, I.R., Veizer, J., and Land, L. S., eds., Stable isotopes in sedimentary geology: Society of Economic Paleontologists and Mineralogists Short Course 10, p. 1.1–1.151.

Cohen, K.M., Finney, S.C., Gibbard, P.L., Fan, J.-X., 2013, The ICS International Chronostratigraphic Chart. Episodes 36 (3), 199-204.

Coplen, T.B., 2007, Calibration of the calcite-water oxygen-isotope geothermometer at Devils Hole, Nevada, a natural laboratory. Geochimica et Cosmochimica Acta 71, 3948-3957.

Mouchi, V., de Rafélis, M., Lartaud, F., Fialin, M., Verrecchia, E., 2013, Chemical labelling of oyster shells used for time-calibrated high-resolution Mg/Ca ratios: A tool for estimation of past seasonal temperature variations. Palaeogeography, Palaeoclimatology, Palaeoecology 373, 66-74.

Pauley, G. B., Van Der Raay, B., Troutt, D., 1988, Species profiles: Life histories and environmental requirements of coastal fishes and invertebrates (Pacific Northwest)ăˇA ˇTPacific oyster, U.S. FishWildl. Serv. Biol. Rep. 82(11.85), U.S. Army Corps of Engineers, TR EL-82.4., Research and Development, National Wetlands Research Center, Washington, DC, 20240, 28 pp.

Ullmann, C.V., Wiechert, U., Korte, C., 2010, Oxygen isotope fluctuations in a modern North Sea oyster (Crassostrea gigas) compared with annual variations in seawater temperature: Implications for palaeoclimate studies. Chemical Geology 277, 160-166.

Ullmann, C.V., Böhm, F., Rickaby, R.E.M., Wiechert, U., Korte, C., 2013, The Giant Pacific Oyster (Crassostrea gigas) as a modern analog for fossil ostreoids: Isotopic (Ca, O, C) and elemental (Mg/Ca ,Sr/Ca, Mn/Ca) proxies. Geochemistry, Geophysics, Geosystems 14 (10), doi: 10.1002/ggge.20257.

Rimstidt, J.D., Balog, A., Webb, J., 1998. Distribution of trace elements between carbonate minerals and aqueous solutions. Geochimica et Cosmochimica Acta 62 (11), 1851-1863.
Wanamaker Jr, A.D., Kreutz, K.J., Wilson, T., Borns Jr, H.W., Introne, D.S., Feindel, S., 2008. Experimentally determined Mg/Ca and Sr/Ca ratios in juvenile bivalve calcite for Mytilus edulis: implications of paleotemperature reconstructions. Geo-Mar Lett 28, 359-368.

Wisshak, M., López Correa, M., Gofas, S., Salas, C., Taviani, M., Jakobsen, J., Freiwald, A., 2009, Shell architecture, element composition, and stable isotope signature of the giant deep-sea oyster Neopycnodonte zibrowii sp. N. from the NE Atlantic. Deep-Sea Research I 56, 374-407.

**Anonymous Referee #2**

The manuscript "An assessment of latest Cretaceous Pycnodonte vesicularis (Lamarck, 1806) shells as records for palaeoseasonality: A multi-proxy investigation" of de Winter and coauthors wants to assess the potential of shells of the bivalve Pycnodonte vesicularis as recorder of palaeoseasonality. They analyzed several specimens coming from the late Maastrichtian Neuquén Basin in Argentina, using different techniques to check the preservation of the shells (CT scanning, light microscopy, Micro- XRF and cathodoluminescence) and to reconstruct the palaeoclimatic and palaeoenvironmental variations recorded by the bivalve (stable isotope, trace

*elements and clumped isotope analyses). They described in great details the methodology used and deeply discussed the advantages and disadvantages of the different methods. The authors discussed in a proper way their results making comparison with recent closely related genera and with data coming from the literature, providing a huge amount of new information.*

*Results are reported in great detail, which causes the manuscript to be very long and often not fluid, due to the wealth of information provided. I understand the need to document and discuss in details the trend observed; however, I think that shorten the manuscript would definitely improve the reading. Part of the method descriptions can be moved to the supplementary material, as well as parts of the comparison with other species should be reduced. Also, the discussion (6.4 temperature proxies) and the conclusions should be shortened, as many times they results in a repetition of the same concepts.*

As mentioned above, we will take this major comment into account in our revision of the manuscript. We aim to shorten the manuscript text by moving parts of the methodology to the supplementary material and to shorten the discussion in the places suggested by Referee #2. The section about temperature proxies (6.4) will be rephrased following comments by Referee #1 and we will aim for a more concise discussion in this paragraph.

*The manuscript address interesting scientific questions that are within the scope of Climate of the Past, so I recommend its publication after moderate revision.*

*Specific comments*

*A) Paragraph 4.1. According to Figure 1, it seems that only one level with Pycnodonte vesicularis is found in the section. The caption specifies that only the main Pycnodonte level is shown. From this, I understand that there are more levels with Pycnodonte but this is not adequately described and clarified in the text. The authors only said that Pycnodonte specimens were collected from the upper 5 m below the Cretaceous- Paleogene boundary. Were the seven specimens analyzed coming from different levels? Some of the differences the authors observed among the specimens may be due to the fact they did not live during the same time interval, thus not experiencing the same environmental oscillations. Also, it is worth to add something about the taphonomic condition of the specimens (e.g., articulated, disarticulated) and the associated fauna, if present.*

As mentioned above, we will adapt paragraph 4.1 and Figure 1 according to the comments by Referee #2 and add information about the stratigraphic constraint on the shells and their state during collection. Furthermore, we will add some information about the abundance of *Pycnodonte* in these strata to Figure 1 to provide context to the discussion of the palaeoenvironment in which these animals lived.

*B) Lines 436-437 and 449. A salinity decrease by fresh water input can also cause the low _18O and _13C values observed, lowering both the _18O and _13C values (fresh water is enriched in 16O and 12C) (Gillikin, 2005; Gillikin et al., 2006). The authors should add a sentence on this and better explain why they excluded the salinity effect. Lines 436-437, add a reference to: "Such a relationship between _18O and _13C has often been interpreted as a sign of diagenetic alteration."*

Both Referees mention the role of salinity in their comments and we will discuss salinity changes in more detail in the revised discussion. We thank Referee #2 for the suggestions on how salinity changes may (partly) explain variations in stable isotope ratios in our records.

*C) Paragraphs 6.3.1-6.3.3 (mainly lines 703-704). When comparing the isotope values of P. vesicularis with related species, the authors have to take in mind the different environmental settings in which the 3 species live (P. vesicularis, N. zibrowii and H. hyotis). As observed by the authors N. zibrowii lives in deep water, so its isotope signatures (especially the _18O values) are also controlled by this parameter. The higher _18O values recorded in N. zibrowii compared to P. vesicularis may be also explained with the deep sea habitat of the former species. So if they want to compare the isotope values, they have to consider species coming from similar environments.*

See above, this is a valid comment and we will mostly restrict our discussion here to the comparison with *Hyotissa* while shortening the discussion of the comparison with *N. zibrowii*, see comment A) from Referee #2.

*D) Lines 709-712. This sentence is strange; are you sure is the juvenile and not the adult part of the shells showing an ontogenetic trend in _13C? Usually bivalves incorporates isotopically light CO2 in the adult stages, showing an ontogenetic decrease in _13C (e.g., Gillikin, 2005; Gillikin et al., 2006, 2007). The model of Lorrain et al. (2004) suggests that the decrease in _13C through ontogeny is actually caused by increasing utilization of metabolic C (respiratory CO2 which is 13C-depleted) to satisfy carbon requirements for calcification. As bivalves grow and become older, the amount of available metabolic CO2 increases, while the amount needed for shell growth is reduced, resulting in more metabolic carbon (12C-enriched) being incorporated into the shell. A similar ontogenetic trend is observed in specimens M6 and M11. The authors should rewrite this part.*

We thank Referee #2 for this comment which will allow us to better discuss carbon isotope ratios measured in our bivalve shells. We will apply the suggestions of the Referee to revise the concerning paragraph of the discussion and provide a better explanation of the stable carbon isotope records plotted in Figure 5 in terms of physiological changes. In addition, as mentioned above, we will limit the comparison with *N. zibrowii* because it is most likely not a very good modern analogue for *N. vesicularis*.

*E) The authors provide a lot of data in the manuscript, analyzing in details the different methods used. I understand that the primary aim of the manuscript is to assess the potential of P. vesicularis shells as recorders of palaeoseasonality. However, the authors obtained some useful data for palaeoclimatic reconstructions which are not adequately discussed in the manuscript. How the data in terms of palaeotemperatures and palaeoseasonality fit into the larger context of the Cretaceous climate of the area? Which new information can they add to the knowledge of the late Cretaceous of the Neuquén basin?*

In the current version of our manuscript, we deliberately kept the palaeoclimate interpretation of our shell records to a minimum, because of the many uncertainties discussed in the manuscript. Some of these uncertainties were picked up in a comment by Referee #1 about our interpretation of the various temperature proxies. We do not want to over-interpret our data sets, but we do agree that some palaeoclimatic context may be a good addition to the manuscript. In the revised version, we will therefore include a short paragraph at the end of the discussion in which tentative palaeoclimate reconstructions from *P. vesicularis* shells are placed in a broader context of palaeoclimate reconstructions in the late Cretaceous.

*Minor comments*

*A) Be consistent through the manuscript on the use of English or American spelling (paleo -> palaeo, recrystallization, recrystallised, : : :)*

We will scrutinize the text and correct inconsistencies in style.

*B) When citing a paper within the manuscript use the same format. Some citations have comma before the year other not, e.g., Kiessling et al. 2005 (line 177) or Woelders et al., 2017 (line 182). Check carefully through the text.*

This will be corrected

*LINE 68-70 Bivalve shells are also important as they have a broad biogeographic distribution, occurring in different environmental settings, from shallow water to deep-sea environments, in freshwater, marine and brackish settings, from near the poles to the equator (e.g., Schöne et al., 2005a)*

We thank Referee #2 for this addition and will implement it in the revised manuscript.

*LINE 78 Add other references as Schöne et al., 2005b; Butler et al., 2013*

These will be added

*LINES 92-96 Add reference to Crippa et al., 2016*

We will add this reference

*LINE 111 ReconstructionS*

This will be corrected

*LINE 130 "The aim of this multi-approach is to characterize the MICROstructure". Refer also in other part of the text to microstructure and not structure, as you are observing shells at micrometrical scale*

This will be corrected

*LINE 196 "from the upper 5 m OF below the Cretaceous". Delete OF*

This will be corrected

*LINE 200 What do you mean by biodegradation? Please explain*

We mean the extent to which the shells suffered from borings of predatory or parasitic organisms (e.g. sponges and polychaete worms). The sentence will be rephrased to clarify this.

*LINE 242 It is not Figure 1, please correct*

This should be Figure 2, we will change the reference

*LINE 244 "See section 4.1.1 and 4.1.3". May it be section 5?*

Indeed, this should refer to sections 5.1.1 and 5.1.3. We will change the references.

*LINES 252 and 274 Gillikin et al., before Lorrain et al.*

This will be corrected

*LINE 257 Surge and Lohmann 2008 before Wanamaker et al. 2008*

This will be corrected

*LINE 294 Add space between 100 and _m*

This will be corrected

*LINE 345 Add reference to MacDonald et al., 2009*

This reference will be added.

*LINE 371 Diagenetic alteration instead of diagenesis*

This will be corrected

*LINES 384-385 and 476- 477 What about the CL of the vesicular layer? Add an image of this; if not in the main paper, add more images in the supplementary. It is important to document what you saw and described.*

We will add more CL images in the supplementary data and add an appropriate image to Figure 3 to refer to in the text.

*LINE 401 Delete space after record*

This will be corrected

*LINES 401-403 Rephrase this sentence*

This sentence will be rephrased with reference to comments from both Referees.

*LINE 405 "In three out of four specimens", delete OUT*

This will be corrected

*LINE 445 What does it mean from the same locality? Same stratigraphic level?*

As mentioned above, we will add more detailed information on the stratigraphic context of the shells in a revised version of the manuscript.

*LINE 446 Defliese et al., 2015; the year should be in parentheses*

This will be corrected

*LINES 471-474 Also, oystreids, due to their layered shell structure, may be more prone to infiltration of fluids inside the shells, which of course affected more the porous chalky fabric than the foliated ones.*

Agreed, we will mention this in the text.

*LINE 477 Laminae instead of lamina*

This will be corrected

*LINE 521 Measuring instead of measured*

This will be corrected

*LINE 566 "in vesicular calcite this close". Delete THIS*

This will be corrected

*LINE 596 ReconstructionS*

This will be corrected

*LINE 611 "vesicular calcite into the in the shell". Delete INTO THE*

This will be corrected

*LINE 651 Klein et al. before Ullmann et al.*

This will be corrected

*LINE 654 Delete "in the Late Cretaceous" at the end of the sentence; it is clear you are referring to the Late Cretaceous*

This will be corrected

*LINE 662 Hyotissa not Hytissa; add the name of the author who first describe the species*

This will be corrected

*LINE 683 "which complicates interpretation"; it should be "which complicates THE interpretation" or "which complicates interpretationS"*

This will be corrected

*LINE 693, 704 and other lines Wisshak et al., 2008, in the reference list is Wisshak et al., 2009*

We will thoroughly check the citations and reference list in the revised version of the manuscript.

*LINE 705 Titschack et al., 2010 before Ullmann et al., 2010*

This will be corrected

*LINE 731 Evaporitic setting*

This will be corrected

*LINE 732 Specify in which country Safaga Bay is*

This will be corrected

*LINE 732 Add + before 2.17 ‰*

This will be corrected

*LINE 734 "records OF H. hyotis". Add OF*

This will be corrected

*LINE 745 "a decrease in salinity in the spring". Delete THE*

This will be corrected

*LINE 781-782 It seems strange that during high productivity spring they growth slower, they should do the opposite. Is there any evidence in previous literature on this?*

A similar point was made by Referee #2 and we will revise this part of the discussion, taking into account the comments from both referees, to better discuss the observed proxy relationships. We acknowledge that the discussion of the phase relationships between proxies may not be clear at this point. We hope that a revision and shortening of this part of the discussion will make the rationale easier to follow.

*LINE 783 Gillikin et al., 2005 before Lorrain et al., 2005*

This will be corrected

*LINE 796 Such a decrease of nearly 10_C between surface and relatively deep sea water is comparable to present day situation?*

We do not wish to argue that this depth temperature profile can explain the full temperature offset (see next line)..

*LINE 801 "d18O", change with _18O*

This will be corrected

*LINE 806-808 During the spring-summer seasons the authors reported a salinity decrease; slow growth may be caused by this?*

We thank Referee #2 for this suggestion and will implement it in a revision of the discussion. This decrease in salinity may indeed cause stress to the bivalve and diminish the growth rate, explaining the observed phase relationships between proxies.

*LINE 815 "than parts of the year". Add OTHER parts of the year*

This will be corrected

*LINE 847 How was 20 PSU determined? It is a very big variations. For example in the Mediterranean Sea a salinity change of 2 PSU would correspond to a shift of _1‰ in _18Osw (Rohling and Bigg, 1998), which is equivalent to nearly 4-5_C in the temperatures calculated from the _18O of the shell. The authors observed a 10_C variation, which correspond to _2-3‰ in _18O. The salinity 20 PSU value seems overestimated. The authors should better explain this assertion*

As mentioned above, we will discuss salinity changes in a more relative sense in the revised version of our manuscript. We realize that this conversion of d18O to psu values is unconventional and not well founded. We will refrain from any quantitative attempt to reconstruct salinity in the revised version, because we cannot constrain all the variables needed for this reconstruction.

*LINE 869 10-15 _C at which water depth?*

These are surface water temperatures. This will be clearly indicated in the text.

*LINE 909 reconstructionS*

This will be corrected

*LINE 911 vesicular instead of vesulicar*

This will be corrected

*Reference list Please check very carefully the reference list. Some data are missing (pages), many specific names are not in italic, some references are in wrong chronological order, some present in the list are missing in the main text and viceversa. Some of the changes to make are listed hereafter:*

The reference list will be thoroughly checked in a revised version and the helpful comments on the formatting and completeness of the references posed by both reviewers will be taken into account.

*Figures*

*FIGURE 1 Is it possible to add a legend with the lithologies? Also, in the y-axis of the log correct BOUNDARY*

This will be corrected, we will add a legend for the lithologies

*FIGURE 2 To be more clear the direction of growth of the shell should be added.*

The red arrow in this figure indicates the direction of growth. This will be explicitly mentioned in the caption.

*FIGURE 3 Images C-G and H, I should be a bit larger. Images C and E are not very clear.*

The figure will be edited according to these suggestions by making the microscope images larger and improving the figure caption.

*FIGURES 5 and 9 Vertical bars have too similar colors (orange and red), change one to be more clear.*

The figure will be edited according to these suggestions

*FIGURE 6 Cross plots showing cross plots, please rephrase.*

The figure caption will be edited according to these suggestions

*FIGURE 8 "interpretation of annual cyclicity based on Sr/Ca ratios" and on _18O seasonality?*

We will add this to the caption of the figure.

*References*

*Butler, P.G., Wanamaker Jr., A.D., Scourse, J.D., Richardson, C.A. & Reynolds, D.J. (2011), Longterm stability of 13C with respect to biological age in the aragonite shell of mature specimens of the bivalve mollusk Arctica islandica, Palaeogeography, Palaeoclimatology, Palaeoecology, 302(1), 21-30.*

*Crippa G., Angiolini L., Bottini C., Erba E., Felletti F., Frigerio C., Hennissen J.A.I., Leng M.J., Petrizzo M.R., Raffi I., Raineri G., Stephenson M.H. (2016). Seasonality fluctuations recorded in fossil bivalves during the early Pleistocene: Implications for climate change. Palaeogeogr. Palaeoclimatol. Palaeoecol., 446, 234–251.*

*Gillikin, D.P. (2005), Geochemistry of Marine Bivalve Shells: the potential for paleoenvironmental reconstruction. Unpublished PhD thesis, Bruxelles University.*

*Gillikin, D.P., Lorrain, A., Bouillon, S., Willenz, P. & Dehairs, F. (2006), Stable carbon isotope composition of Mytilus edulis shells: Relation to metabolism, salinity, _13CDIC, and phytoplankton, J. Organic Geochemistry, 37, 1371–1382.*

*Gillikin, D.P., Lorrain, A., Meng, L. & Dehairs, F. (2007), A large metabolic carbon contribution to the _13C record in marine aragonitic bivalve shells, Geochimica et Cosmochimica Acta, 71, 2936–2946.*

*Lorrain, A., Paulet, Y.M., Chauvaud, L., Dunbar, R., Mucciarone, D. & Fontugne, M. (2004), _13C variation in scallop shells: increasing metabolic carbon contribution with body size?, Geochimica et Cosmochimica Acta, 68, 3509–3519.*

*MacDonald, J., Freer, A., & Cusack, M. (2009). Alignment of crystallographic c-axis throughout the four distinct microstructural layers of the oyster Crassostrea gigas. Crystal Growth & Design, 10(3), 1243-1246.*

*Rohling, E.J., Bigg, G.R. (1998). Paleosalinity and _18O: a critical assessment. J. Geophys. Res. Oceans 103 (C1), 1307–1318.*

*Schöne, B.R., Fiebig, J., Pfeiffer, M., Gle_, R., Hickson, J., Johnson, A.L.A., Dreyer, W. & Oschmann, W. (2005a), Climate records from a bivalved Methuselah (Arctica islandica, Mollusca; Iceland), Palaeogeography, Palaeoclimatology, Palaeoecology, 228(1), 130-148.*

*Schöne, B.R., Pfeiffer, M., Pohlmann, T. & Siegismund, F. (2005b), A seasonally resolved bottom water temperature record for the period of AD 1866–2002 based on shells of Arctica islandica (Mollusca, North Sea), Int. J. Climatol., 25, 947–962*